# Parasitoid vectors a plant pathogen, potentially diminishing the benefits it confers as a biological control agent

Chang-Fei Guo[1,2,3,6], Muhammad Z. Ahmed [4,6], Da Ou[1,3], Li-He Zhang[1,2], Zi-Tong Lu[1,2], Wen Sang[1,2,3], Cindy L. McKenzie[4], Robert G. Shatters Jr[4] & Bao-Li Qiu [1,2,3,5✉]

Huanglongbing (HLB) is a destructive disease of citrus primarily transmitted by the Asian citrus psyllid (ACP). Biocontrol of ACP is an environmentally sustainable alternative to chemicals. However, the risk of parasitoid rational application in ACP biocontrol has never been evaluated. Here we show, the dominant parasitoid of ACP, *Tamarixia radiata*, can acquire the HLB pathogen *Candidatus* Liberibacter asiaticus (CLas) and transmit it horizontally when probing ACP nymphs. If these ACP nymphs survive the probing, develop to adults and move to healthy plants, CLas can be transmitted to citrus leaves during feeding. We illustrate the formerly unrecognized risk that a parasitoid can potentially serve as a phoretic vector of the pathogen transmitted by its host, thus potentially diminishing some of the benefits it confers via biocontrol. Our findings present a significant caution to the strategy of using parasitoids in orchards with different infection status of insect-vectored pathogens.

[1] Key Laboratory of Bio-Pesticide Innovation and Application of Guangdong Province, South China Agricultural University, Guangzhou 510640, China. [2] Guangdong Laboratory for Lingnan Modern Agriculture, Guangzhou 510640, China. [3] Engineering Research Center of Biocontrol, Ministry of Education, Guangzhou 510640, China. [4] Subtropical Insects and Horticulture Research Unit, Agricultural Research Service, USDA, Fort Pierce, FL 34945, USA. [5] College of Life Sciences, Chongqing Normal University, Chongqing 401300, China. [6] These authors contributed equally: Chang-Fei Guo, Muhammad Z. Ahmed. ✉email: baileyqiu@scau.edu.cn

Citrus Huanglongbing (HLB) or citrus greening has become one of the world's most destructive diseases since its global spread during the last 20 years. It was first reported from southern China in 1919 and now is known to occur in approximately 40 different countries across Asia, Africa, Oceania, South and North America[1,2]. It has been estimated to have cost Florida's economy over US$4.4 billion and resulted in over 8257 job losses since 2006[3]. The causative agents of HLB are three vector-borne α-proteobacteria, of which the most widespread and pervasive is *Candidatus* Liberibacter asiaticus (CLas), vectored by the Asian citrus psyllid (ACP), *Diaphorina citri* Kuwayama[1]. Therefore, effective control of ACP is a key component of CLas management globally.

Chemical control has been used aggressively to eliminate ACP from citrus orchards in many countries. However, the frequent use of insecticides has resulted in the evolution of insecticide resistance for ACP in many parts of the world, caused a decrease in the populations of ACP natural enemies, and raised concerns regarding human health and environmental sustainability as well as citrus production costs[4–6]. Thus, increasing effort has been placed on developing biological control solutions to ACP. Among the biological control agents of ACP, *Tamarixia radiata* (Waterston) is considered the dominant parasitoid globally[7,8]. For example, the parasitism of *T. radiata* on ACP was approximately 36% in summer and 46% during autumn in China[9]. It was observed to vary between 20 and 56% in Florida[10], and averaged around 21% in southern California in autumn[11]. Pluke et al.[12] also observed between 70 and 100% parasitism in Puerto Rico. *T. radiata* is a solitary, arrhenotokous ectoparasitoid, adult females of *T. radiata* feed on all the nymphal instars; however, they generally oviposit on the older nymphal instars (preferentially fourth and fifth instars)[13,14]. This parasitoid has been used for the biological control of ACP in many regions of the world, and one field investigation has successfully demonstrated suppression in the ACP population in the natural environment without applying insecticides[12].

Prior studies have revealed two main routes for the increasing distribution of the CLas pathogen; grafting and ACP feeding between CLas infected and healthy but susceptible citrus plants. However, during biological control of ACP using *T. radiata*, we unexpectedly found that *T. radiata* can acquire CLas as nymphs and subsequent adults can transmit it to ACP nymphs. This observation raises the concern that such transmission of CLas by *T. radiata* could diminish the benefits of biological. In the current study, we evaluated the acquisition, persistence, and transmission of CLas pathogen in the *T. radiata*-ACP-citrus plant tritrophic system. We present a new insight into parasitoid-based biological control of ACP and reveal a new route of CLas transmission. The discovery that *Tamarixia* parasitoid can function as a vector of CLas creates an academic need to understand this impact on CLas transmission in the field. *Tamarixia* parasitoid is used in biological control to reduce the ACP population and thus reduce the transmission of CLas from infected to healthy citrus. The question now is raised: is the increased percentage of the ACP population that is infected with CLas, as a result of *Tamarixia* parasitoid transmission, significant enough to outweigh the benefit achieved by ACP population reduction through *Tamarixia* parasitism? Also, our findings demonstrate that the potential negative effect of parasitoid-based horizontal transmission of plant pathogens should carefully appraise in other tritrophic systems such as plant-aphid/whitefly/mealybug-parasitoid interactions.

## Results

### PCR analysis of acquisition and persistence of CLas in *Tamarixia radiata*.
When *T. radiata* eggs hatch inside CLas infected psyllids, the CLas titers generally increased with the development of *T. radiata* immatures. As expected, CLas was not detected in eggs. CLas titers in the 3rd and 4th instar larvae were significantly higher than those of 1st or 2nd instar larvae, pupae, and adults (Fig. 1a and Supplementary Data 1). Interestingly, the CLas titer of males was significantly higher than that of females (Fig. 1b and Supplementary Data 1). As time went by, CLas titers in *T. radiata* female adults decreased gradually to undetectable levels and totally disappeared from qPCR detection on the 10th day after emergence (Fig. 1c and Supplementary Data 1).

Further investigation demonstrated that CLas was detectable in different tissues of *T. radiata* adults, but their titers varied among various tissues, for example, the highest titer was in hemolymph, followed by the gut, fat body, ovary, salivary glands, and spermatheca. It was noteworthy that CLas was also found in the poison sac and chest muscles of *T. radiata* (Fig. 1d and Supplementary Data 1).

### FISH Localization of CLas in different stages of *Tamarixia radiata*.
CLas infections in *T. radiata* were localized visually using fluorescence in situ hybridization. CLas fluorescence was visually confirmed in both the poison sac and ovipositor of female *T. radiata* and in all the larval stages of the $F_0$ generation, but not in the eggs (Fig. 2a−f), which was consistent with the qPCR detection results (Fig. 1). Over the course of development of *T. radiata* larvae, CLas extended its distribution gradually, and as a result, CLas had the strongest fluorescent signal in the middle of the abdomen of 4th instar larvae (Fig. 2e). In addition, for the FISH examined 1st to 4th instar larvae of *T. radiata*, 15/20, 17/20, 17/19, and 16/17 were CLas infected respectively, and all the infected individuals in each instar stage were consistent to each other with their CLas distributions.

### FISH Location of CLas in different organs of *Tamarixia radiata*.
CLas bacteria could be FISH visualized in the gut, salivary glands, muscle, and fat body (Fig. 3a−d and Supplementary Fig. 1a−d); CLas was also localized throughout the tissues of female ovaries and male spermatheca (Fig. 3e, f and Supplementary Fig. 2a, b). In addition, CLas had a scattered localization pattern throughout the poison sac (Fig. 3g and Supplementary Fig. 2c) and the DuFour's gland (Fig. 3h and Supplementary Fig. 2d).

### Maternal transmission of CLas between *Tamarixia* generations.
The relative titers of CLas in different stages of *T. radiata* $F_1$ progeny were examined using qPCR. The titer of CLas in eggs was significantly higher than that in other life stages; it gradually reduced with larvae development. Eventually, CLas was undetectable in the pupal and adult stages of *T. radiata* $F_1$ progeny (Fig. 4 and Supplementary Data 1), which indicated that *T. radiata* could not successfully transmit CLas vertically to its $F_2$ generation.

FISH visualization showed strong CLas fluorescence signals in $F_1$ *T. radiata* eggs exhibiting a scattered distribution pattern (Fig. 5a). With the development of *T. radiata* $F_1$ progeny, the fluorescence signal of CLas significantly weakened until it almost disappeared in pupal and adult stages (Fig. 5b−e). The FISH visualization results were consistent with the qPCR detection results.

### Localization of CLas in ACP following inoculation via parasitoid probes.
ACP nymphs probed by a CLas-infected parasitoid at their 4th instar stage (hereafter referred to as *T. radiata*-inoculated ACP) only had CLas detected in their hemolymph, but was not found in the salivary glands or midgut when they developed to 5th instar nymphs (Fig. 6). However, CLas was

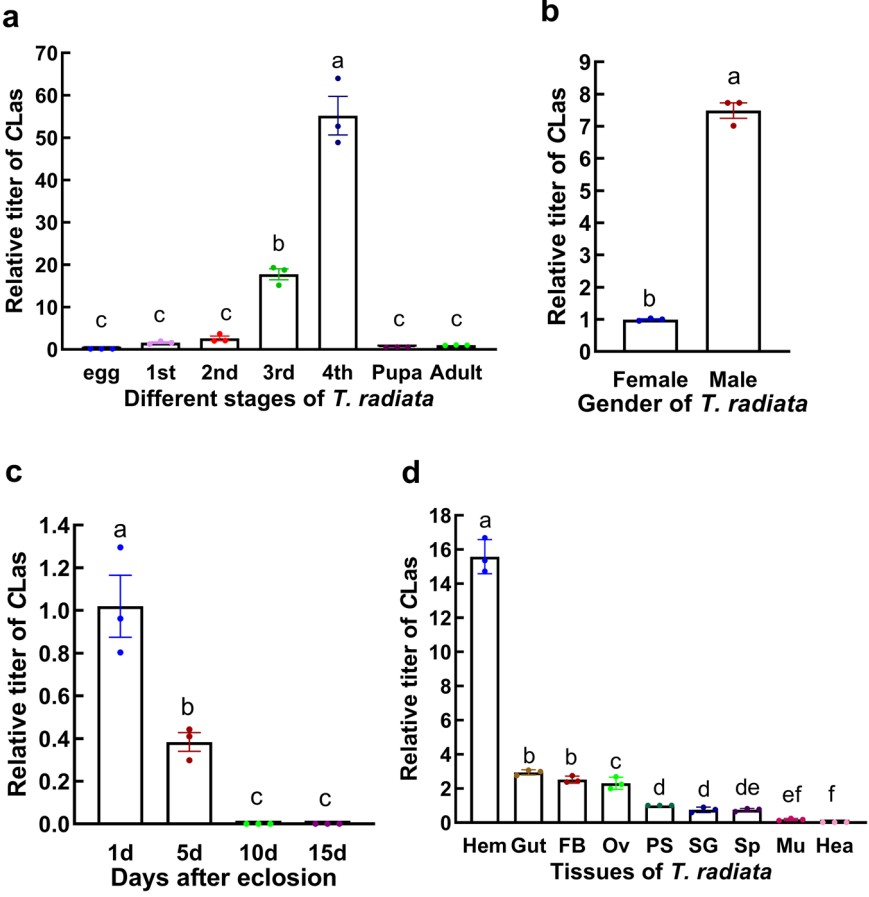

**Fig. 1 Acquisition and persistence of CLas in F$_O$ Tamarixia radiata developed from CLas donor ACP hosts. a** Relative titers of CLas in different developmental stages of F$_O$ *T. radiata* parasitoids; **b** relative titers of CLas in male and female adults of F$_O$ *T. radiata*; **c** the retention time of CLas in the female adults of F$_O$ *T. radiata*; **d** relative titers of CLas in different tissue of newly emerged *T. radiata*, Hem: hemolymph, Gut: gut, FB: fat body, Ov: ovary, PS: poison sac, SG: salivary glands, Sp: spermatheca, Mu: chest muscle, Hea: head. The titer of CLas was calculated using the method of $2^{-\Delta\Delta ct}$. Error bars represent the mean value ±SE of three replicates Columns with the same letter represent means with no significant difference at $P < 0.05$.

detected both in the hemolymph and salivary glands in 8-day old adults, but still not in their midgut. The CLas titer in adults' hemolymph was significantly higher than that in 5th instar nymphs due to its proliferation. All these findings suggest that after inoculation by *T. radiata*, CLas titer accumulates in the hemolymph of 5th instar ACP nymphs, as well as in the hemolymph and salivary glands of 8-day old ACP adults, but fails to enter their midguts (Fig. 6 and Supplementary Data 1).

FISH was used to confirm the findings of qPCR. In ACP adults that acquired CLas directly from citrus plants, CLas was clearly visible in midguts (Fig. 7c and Supplementary Fig. 3c); however, no CLas FISH signal was found in the midgut of 5th instar nymphs (Fig. 7a and Supplementary Fig. 3a) or 8-day old adults (Fig. 7b and Supplementary Fig. 3b) of the *T. radiata*-inoculated ACP. In addition, CLas was not visualized in the salivary glands of these 5th instar ACP nymphs (Fig. 7d and Supplementary Fig. 4a). The CLas was visualized in the central area of salivary glands of the 8-day old *T. radiate*-inoculated ACP adults (Fig. 7e and Supplementary Fig. 4b) and when using salivary glands of plant-inoculated ACP adults (Fig. 7 f and Supplementary Fig. 4c). Our findings suggest that CLas in *T. radiata*-inoculated ACP accumulates and proliferates during the development of its ACP host. The FISH results were consistent with that of the qPCR results.

**CLas transmission to citrus plants by *T. radiata*-inoculated ACP.** After 30 days of feeding, CLas was not detected in the citrus

plants fed on by *T. radiata*-inoculated ACP adults, but was detected in citrus plants fed on by ACP adults that acquired CLas from plants (positive control). After 40 days of feeding by *T. radiata*-inoculated ACP, approximately one-third of the citrus leaves were CLas positive, compared to 100% of citrus leaves fed on by ACP that acquired CLas from plants. However, after 50 days, 100% of the former also tested positive (Supplementary Fig. 5 and Supplementary Data 1). These results demonstrated that CLas can proliferate within citrus plants following infection by ACP adults inoculated by *T. radiata* probing.

**Localization of CLas in citrus plants fed on by *T. radiata*-inoculated ACP.** After plants were fed on by the *T. radiata*-inoculated ACP adults, CLas was mainly distributed in the phloem of the citrus leaf veins (Fig. 8a−c). The rankings of fluorescence signals of CLas based on visual observations among different citrus leaf samples were: leaf from the CLas infected tree (Fig. 8a) >leaf that had been fed on by ACP adults that acquired CLas from plants (positive control, Fig. 8b) >leaf that had been fed on by the *T. radiata*-inoculated ACP adults (Fig. 8c). There was no fluorescence signal in the leaf that had been fed on by the CLas-free ACP adults (negative control) (Fig. 8d).

**Phylogenetic analysis of CLas in different ACP populations and citrus plants.** The CLas bacteria had high fidelity during its horizontal transmission from donor ACP to parasitoid, from

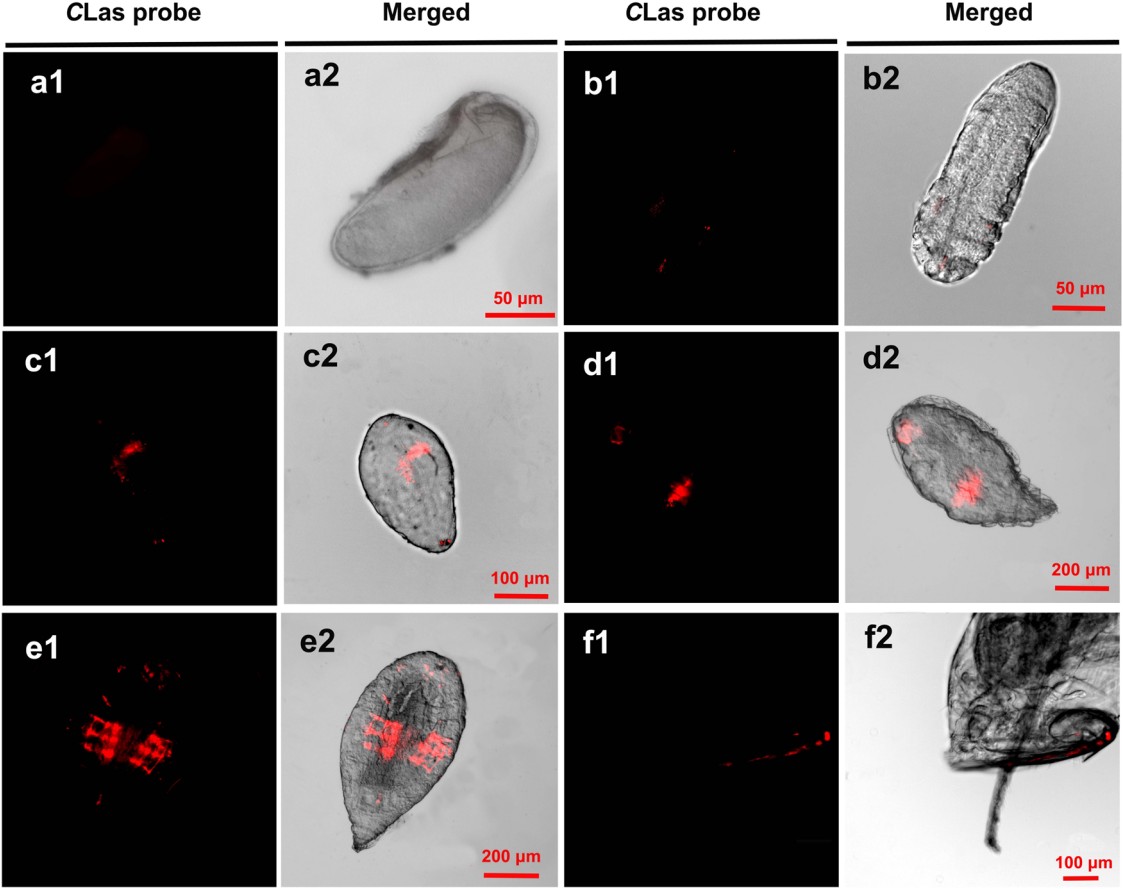

**Fig. 2 The distribution patterns of CLas in F_0 *Tamarixia radiata*.** **a1**/**a2**: egg; **b1**/**b2**: 1st instar; **c1**/**c2**: 2nd instar; **d1**/**d2**: 3rd instar; **e1**/**e2**: 4th instar; **f1**/**f2**: the abdominal poisonous sac and ovipositor of female *T. radiata*. Panels **a1**–**f1** show the CLas probe (red) in a dark field and **a2**–**f2** show merged images.

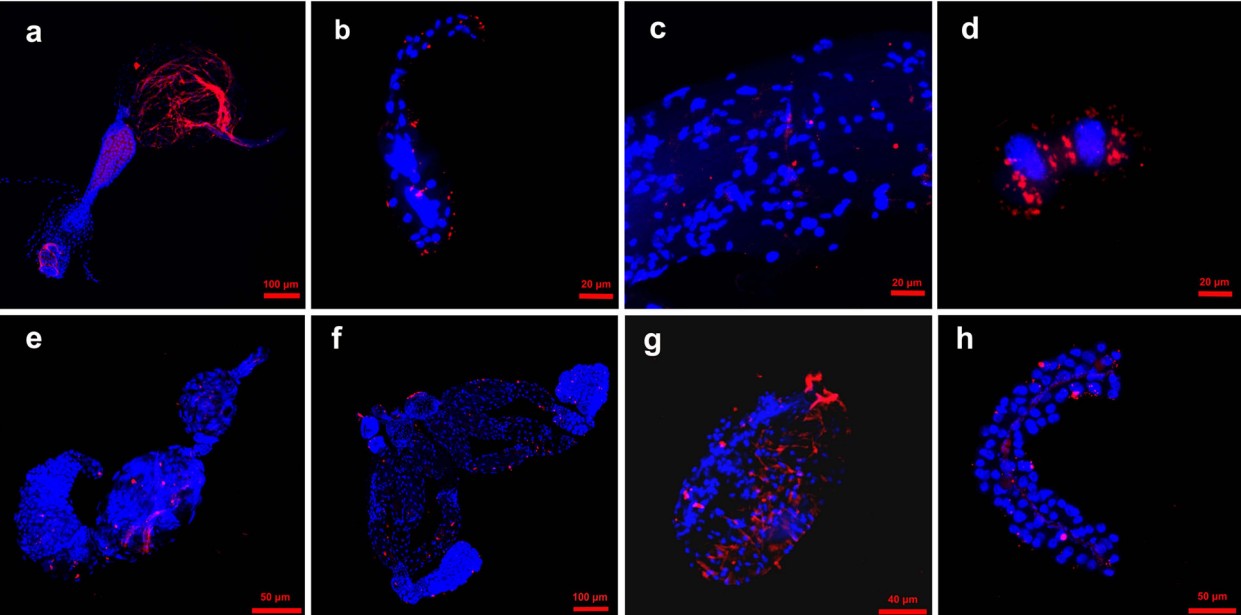

**Fig. 3 Visualization of CLas in different tissues of F_0 *Tamarixia radiata*.** **a** gut; **b** salivary glands; **c** chest muscle; **d** fat body; **e** ovary; **f** spermatheca; **g** poison sac; **h** DuFour's gland. The nuclei were stained with DAPI (blue) and CLas were stained with CLas probe (red).

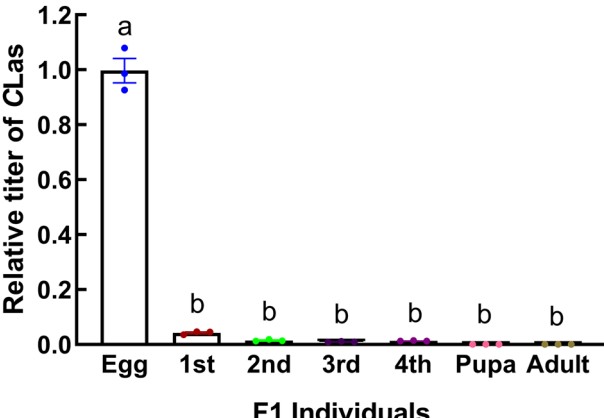

**Fig. 4 The relative titers of CLas in the *Tamarixia radiata* F₁ generation.** The relative titers were calculated using the method of $2^{-\Delta\Delta ct}$. Error bars represent the mean value ±SE. Columns with the same letter represent means with no significant difference at $P < 0.05$. Three replicates of each qPCR were investigated in each column.

parasitoid to *T. radiata*-inoculated ACP and from *T. radiata*-inoculated ACP to recipient citrus plant. There was no variation in their *omp* gene sequences (GenBank accession numbers: MT191373-MT191376). Based on their *omp* gene sequences, all of the CLas isolates grouped into one branch belonging to the Asia *Candidatus* Liberibacter asiaticus (Fig. 9).

## Discussion

The current study demonstrates, for the first time to the best of our knowledge, that the parasitoid *T. radiata* can act as a vector of CLas. It has been proven that CLas pathogen can localize in different tissues and organs of *T. radiata*, after it is initially picked up during its development in CLas-infected ACP hosts. The CLas could be detected by qPCR for at least 5 days in *T. radiata* adult females following their emergence from CLas-infected ACP nymphs. This is similar to findings in a previous study by Ahmed et al.[15] that the bacterial endosymbiont, *Wolbachia*, could be detected in the whitefly parasitoid adult female up to at least 5 days after they had fed on, or probe checked, *Bemisia tabaci* nymphs that were infected with *Wolbachia*.

Before this study, parasitoids, and mites had been found to transmit bacterial endosymbionts horizontally in some previous studies. For example, Jaenike et al.[16] found that the ectoparasitic mites, *Macrocheles subbadius*, feeding on insect hemolymph, can pick up the bacterial endosymbiont *Spiroplasma* species from infected *Drosophila nebulosa* females and horizontally transmit it to *Drosophila willistoni*. Progeny of the recipient *D. willistoni* were infected, indicating successful maternal transmission of *Spiroplasma* into a new host species. Gehrer and Vorburger[17] revealed that the parasitoid *Aphidius colemani* can transmit the bacterial endosymbiont *Hamiltonella defensa* by sequentially stabbing infected and uninfected individuals of their aphid hosts *Aphis fabae*, thus establishing new, heritable infective lines. In addition, our previous study[15] reported that the parasitoid *Eretmocerus* sp. nr. *furuhashii* can phoretically acquire the bacterial endosymbiont *Wolbachia* when probe checking infected whitefly hosts. They can then transmit it to uninfected whitefly hosts horizontally. All previous parasitoid-based horizontal transmissions of bacteria were investigated in endosymbionts only. Our study has revealed, for the first time to the best of our knowledge, the unexpected role of a *Tamarixia* parasitoid in transmitting a plant bacterial pathogen and raises concern about the potential for adverse effects on ACP biological control as it relates to limiting the spread of HLB to uninfected citrus. In addition, Ahmed et al.[15] revealed that both the probing and feeding of *Eretmocerus* sp. nr.

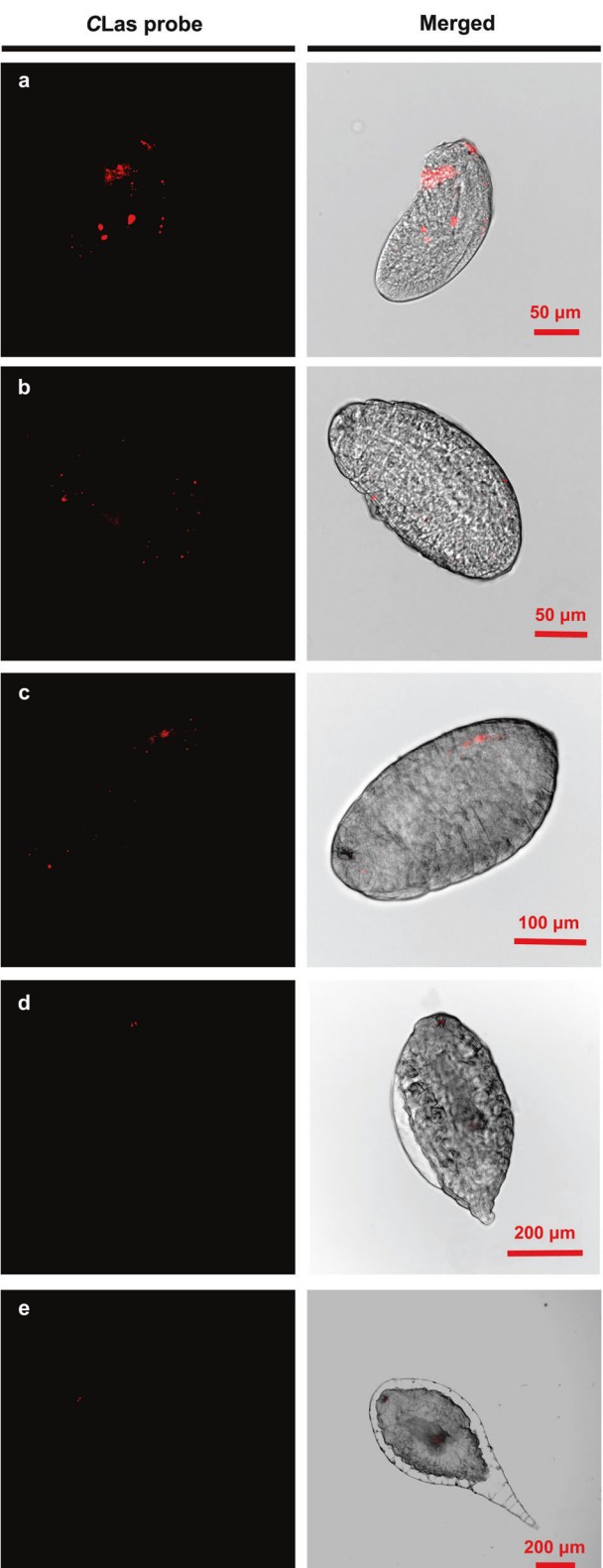

**Fig. 5 The distribution of CLas in the immatures of F₁ *Tamarixia radiata*. a** egg; **b** 1st instar; **c** 2nd instar; **d** 3rd instar; **e** 4th instar. Panels in the left column show the CLas probe (red) in a dark field and in the right column show merged images.

*furuhashii* can result in the horizontal transmission of *Wolbachia* between whitefly hosts, whereas only the female ACP nymphs that were probed by *T. radiata* and survived were investigated in the

current study. If the male and female ACP nymphs that survived *T. radiata* probing are also taken into consideration, the unexpected risk of *T. radiata* vectoring *C*Las transmission may increase, since we have also shown that the *C*Las titer in male *T. radiata* was higher than even in females.

In other studies, its been shown that endosymbiotic bacteria transmit horizontally with poor fidelity[18–20] or fail to persist[21] between different species. Our previous study revealed *Rickettsia* kept 100% fidelity when it was transmitted horizontally between the whitefly *Bemisia tabaci* MEAM1 and MED populations via cotton plants[22]. In agreement with the previous findings[22], the horizontal transmission of *C*Las vectored by *T. radiata* in the current study showed 100% fidelity between the *C*Las bacteria in

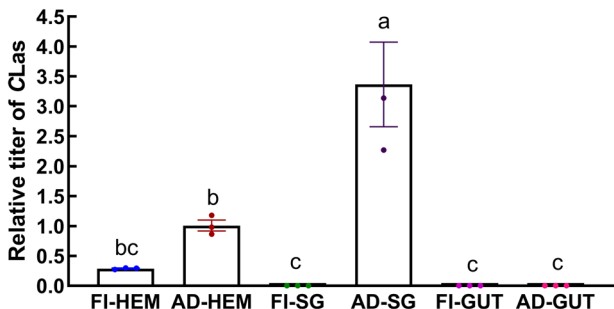

**Fig. 6 The relative titers of *C*Las in the 5th instar nymphs and adults of *Tamarixia radiata*-inoculated Asian citrus psyllid.** FI-HEM: hemolymph of 5th instar nymphs; AD-HEM: hemolymph of adults; FI-SG: salivary glands of 5th instar nymphs; AD-SG: salivary glands of adults; FI-GUT: midgut of 5th instar nymphs; AD-GUT: midgut of adults. The relative titers were calculated using the method of $2^{-\Delta\Delta ct}$ with the house-keeping gene, *β-actin* gene. Error bars represent the mean value ±SE. Columns with the same letter represent means with no significant difference at $P < 0.05$. Three replicates of each qPCR were investigated in each column.

donor ACP, *T. radiata* parasitoids, *T. radiata*-inoculated ACP, and recipient citrus plants.

It is commonly believed that the alimentary canal and salivary glands are the key barriers for the transmission and spread of bacterial pathogens from insect vectors to plants[23–26]. Bacteria usually need to penetrate and accumulate in specific cells of the salivary glands before being successfully transmitted into plants through the salivary glands of insect hosts[27]. Wu et al.[28] reported that *C*Las multiplication was detected in the hemolymph and salivary glands of ACP adults when the bacterium was acquired by ACP nymphs. In conjunction with the previous study, our current study also revealed that *C*Las could multiply in the hemolymph and salivary glands of the ACP adults that developed from nymphs which were probed by *C*Las-vectored *T. radiata* parasitoids. The presence of *C*Las in the citrus leaves fed on by *T. radiata*-inoculated ACP suggests that the parasitoid *T. radiata* can be a potential vector, transmitting *C*Las from the infected to the healthy citrus plants through acquisition from *C*Las infected ACP and then transmission to uninfected ACP that then feed on citrus plants.

When releasing *T. radiata* parasitoids to control ACP in orchards, the *C*Las infection status of ACP and the citrus plants should be determined. This is because *T. radiata* parasitoids, which have fed upon or parasitized the *C*Las-infected ACP, can transmit *C*Las to other ACP at new locations and therefore introduce *C*Las to healthy plants in the orchard. Even if *T. radiata* parasitoids are reared on *C*Las-free ACP and *C*Las-free plants, then released into the orchard, where there are *C*Las-infected ACP and trees that are either *C*Las-free or asymptomatic with undetectable or unknown *C*Las titer, *C*Las can still be picked up and transmitted. Since *C*Las can spread throughout an entire orchard within a year before any disease symptoms appear, according to the study of Lee et al.[29], if there are a few *C*Las-infected asymptomatic trees in the orchard, or if a few *C*Las-vectored *T. radiata* parasitoid adults accidentally get introduced into a *C*Las-free orchard, then there might be a chance that *C*Las invasion could be initiated. It should be noted that our current experiments were performed under laboratory conditions and may not be

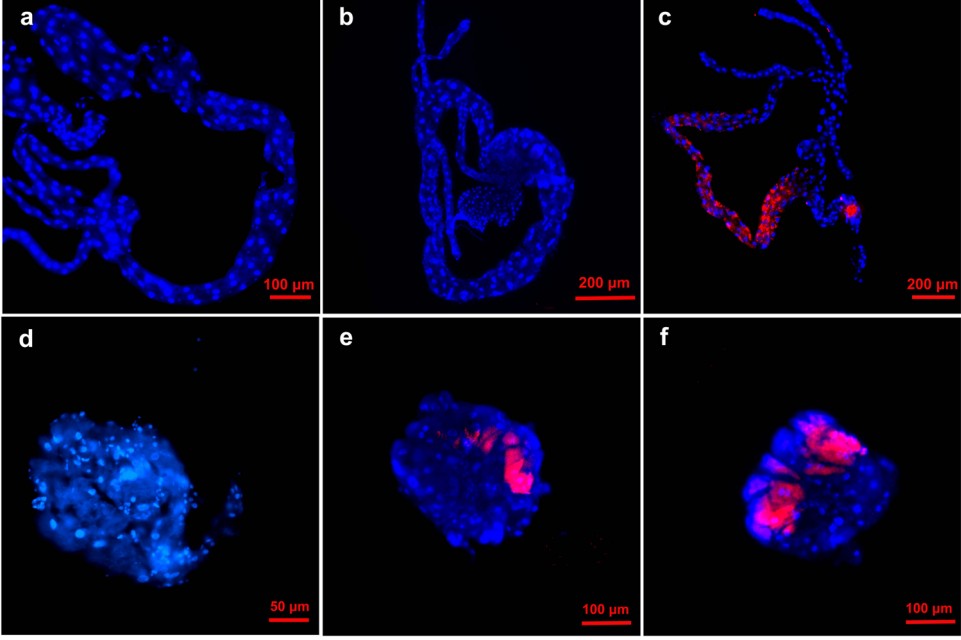

**Fig. 7 FISH visualization of *C*Las in the midgut and salivary glands of *C*Las-recipient Asian citrus psyllid.** **a** midgut of 5th instar ACP nymphs (*C*Las acquired from *T. radiata*); **b** midgut of 8-day old ACP adults (*C*Las acquired from *T. radiata)*; **c** midgut of *C*Las positive ACP adults (*C*Las acquired from citrus plants). **d** salivary glands of 5th instar ACP nymphs (*C*Las acquired from *T. radiata*); **e** salivary glands of 8-day old ACP adults (*C*Las acquired from *T. radiata*); **f** salivary glands of *C*Las positive ACP adults (*C*Las acquired from citrus plants). The nuclei were stained with DAPI (blue) and *C*Las were stained with *C*Las probe (red).

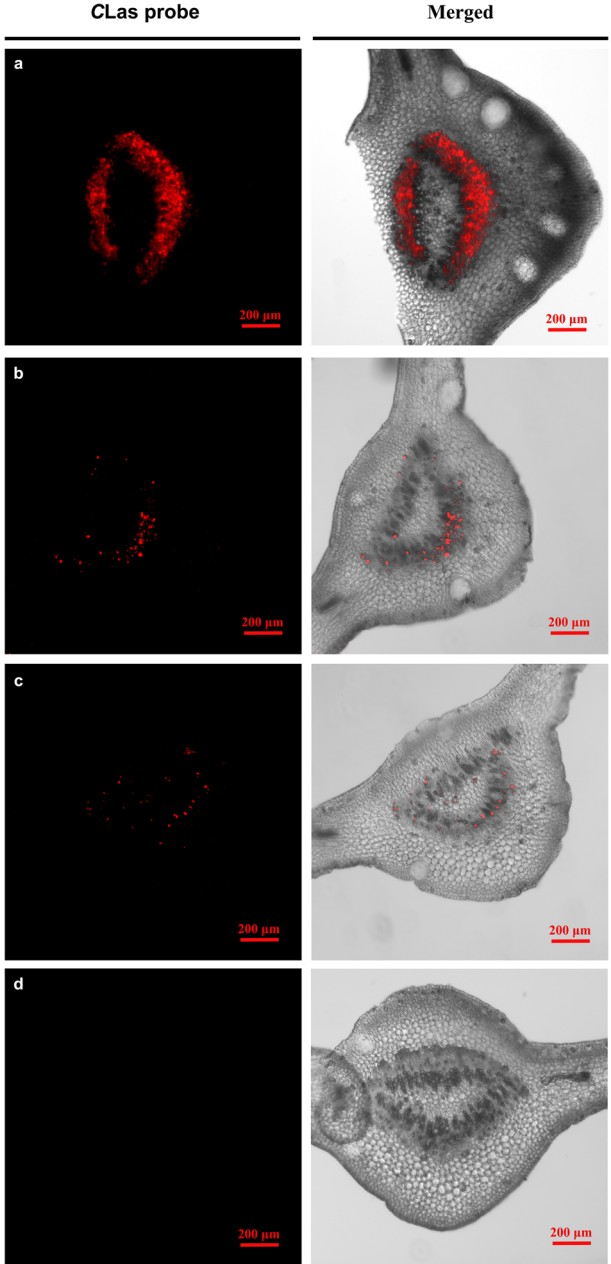

**Fig. 8 Localization of CLas in the recipient citrus plants fed on by Tamarixia radiata-inoculated ACP. a** CLas in leaf veins of CLas-infected plant; **b** CLas in leaf veins that had been fed on by ACP adults that acquired CLas from plants; **c** CLas in leaf veins that had been fed on by *T. radiata*-inoculated ACP; **d** leaf veins of the plants fed on by healthy ACP.

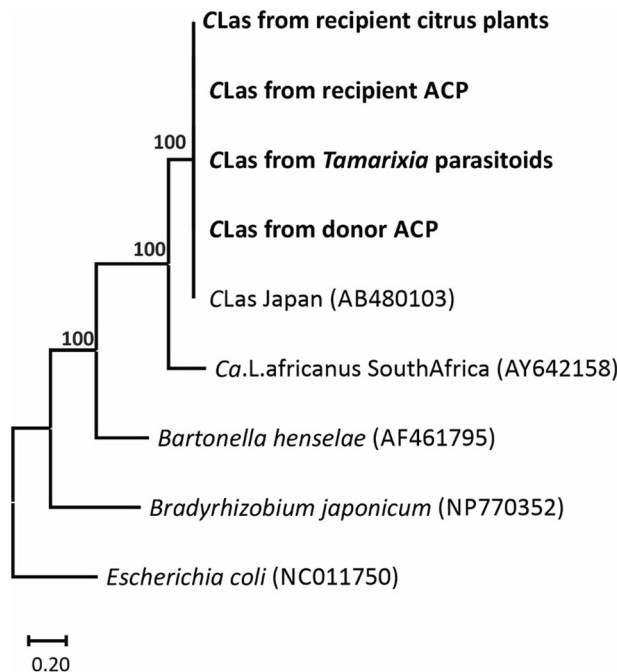

**Fig. 9 Identification of the CLas in different ACP populations and citrus plants.** The maximum likelihood phylogenetic tree was generated based on the alignment of *omp* gene sequences of CLas and other α-proteobacteria, with 1000 non-parametric bootstrap replications in RAxML. *Escherichia coli* was added as an outgroup.

representative of the full suite of biotic and abiotic factors present in field conditions. Our main study revealed that parasitoid vector *T. radiata* can contract and potentially transmit plant pathogen CLas, which will potentially diminishing the benefits it confers as biological control agent. Future research should be conducted using agent-based models to determine the impact of this new mode of transmission in HLB spread and to evaluate the potential risk of using *T. radiata* as a biological agent against ACP in citrus orchards.

In conclusion, our current study reveals that CLas can be picked up by the parasitoid *T. radiata* during their development in CLas-infected ACP nymphs, and that it can persist in the parasitoid adult for at least 5 days following wasp emergence. During its persistence in the parasitoid, this bacterium can be

transmitted to a new ACP host by the wasp feeding or probe checking for oviposition. Eventually, the *Tamarixia*-inoculated ACP can transmit CLas to healthy citrus plants during their feeding (Fig. 10). To our knowledge, this is the first report of an insect natural enemy, actively applied for the biological control of a given pest, could potentially contribute to transmission of a plant pathogen and therefore spread of the resulting plant disease.

## Materials and methods

**Insect rearing.** A CLas negative colony of ACP was initially collected from CLas-free *Murraya exotica* L. growing in the ornamental landscape of South China Agricultural University (SCAU, Guangzhou, China) in May 2014. Then it was reared on potted *M. exotica* in a greenhouse at SCAU. *M. exotica* plants were pruned regularly to promote the growth flushes necessary to stimulate ACP oviposition. The ACP populations were periodically (at least once a month) tested to ensure the colony was CLas-free using nested quantitative PCR detection according to the method described by Coy et al.[30].

The parasitoid *T. radiata* used in the current study was initially collected from ACP hosts on *M. exotica* plants in the above-mentioned location during June 2015. Its population was maintained in rearing cages (60 × 60 × 60 cm) using a CLas-free ACP-*M. exotica* rearing system under laboratory conditions (26 ± 1 °C, RH 80 ± 10% with L:D = 14:10 photoperiods in insect incubators).

**Host plants.** CLas-free and CLas-infected plants of *Citrus reticulata* Blanco cv. Shatangju were used in the current study. Both plant types were obtained from The Citrus Research Institute of Zhaoqing University (Guangdong, China). The CLas-infected plants were inoculated by shoot grafting. All plants were approximately 4-year old and 1.2−1.5 m in height, separated in nylon net greenhouses (70 mesh per inch2) at two different locations about 2.2 km apart in SCAU. Again, nested qPCR detection was performed periodically (at least once a month) to test for the presence or absence of CLas in the citrus plants according to the method described by Coy et al.[30].

**Acquisition and persistence of CLas in Tamarixia radiata.** When new shoots of CLas-infected *C. reticulata* plants were grown to 5–8 cm, 20 pairs of 1 week-old ACP adults were introduced into one nylon bag covering one fresh shoot to lay eggs for 48 h. When the progeny of ACP developed through to 4th or 5th instar nymph (CLas-donor ACP), which are the stages preferred by *T. radiata* parasitoids, 150 of the ACP nymphs were randomly selected and the remaining ones were removed. Following this, 10 pairs of 3-day old *T. radiata* adults, randomly selected from the population that has

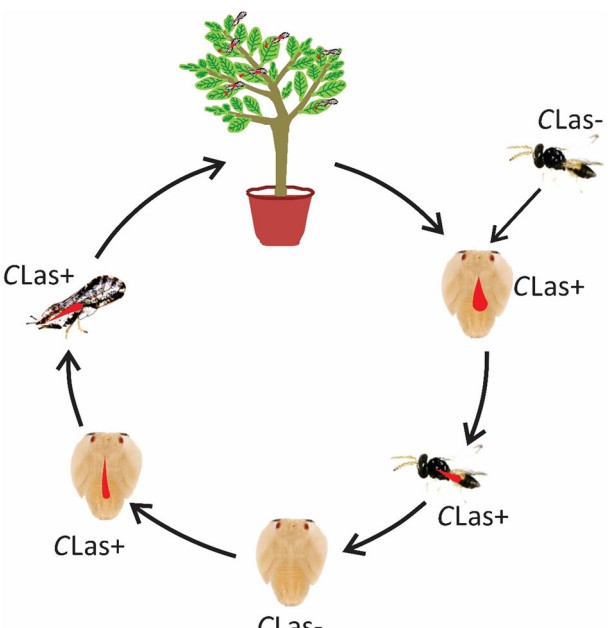

**Fig. 10 Schematic overview of CLas transmission vectored by *Tamarixia radiata* parasitoid.** When ACP nymphs feeding on the CLas positive citrus plants, its parasitoid *Tamarixia radiata* females can acquire CLas from these CLas-infected ACP nymphs and carry it in their ovipositors. After probe checking CLas negative ACP hosts with ovipositor, *Tamarixia* can transinfect CLas into the hemolymph of probed ACP. If parasitized ACP nymphs survive from the parasitizing check, CLas can spread into the salivary glands of the ACP host after proliferation. When the CLas-recipient ACP individuals develop into adults and move to healthy citrus plants, CLas can be transmitted to citrus leaves during recipient ACP feeding, completing the transmission circle of the pathogen from CLas-infected citrus to ACP, to parasitoid, from parasitoid to CLas uninfected-ACP and from CLas-recipient ACP to-citrus plant.

been tested to be CLas-free, were introduced into the nylon bag in order to parasitize the 4th or 5th instar ACP nymphs for 48 h before being recaptured. Then the potentially parasitized ACP nymphs together with the citrus plants were cultured in a plant growth chamber (Jiangnan Instrument Company, RXZ-500D, at 26 ± 1 °C, 60 ± 2% RH and 14:10 h L:D photoperiod of 3,000 lx illumination).

When the progeny of *T. radiata* (considered $F_0$ generation) developed to 3-day egg, 1st to 4th instar larvae, pupae, and adult stages respectively, they were identified and collected with the assistance of a stereomicroscope. DNA of each stage sample was extracted using the TIANamp Genomic DNA Kit (TIANGEN, Beijing, China) for CLas qPCR detection and titer quantification. Thirty eggs, 20 individuals of 1st or 2nd instar, 10 individuals of 3rd or 4th instar larvae or pupa, as well as three individuals of female or male adults were subsequently ground together to represent each life stage in qPCR, and each stage qPCR detection was repeated three times.

The primers used for CLas qPCR detection were LJ900 primers, (F5′-GCCGT TTTAAC ACAAAAGATGAATATC-3′, R5′-ATAAATCAATTTGTTCTAGTTT AC GAC-3′), and 18S rRNA gene of *T. radiata* (F5′-AAACGGCTACCACATCCA-3′, R5′-ACCAGACT TGCCCTC CA-3′)[31] was used as an internal control for DNA normalization and quantification. In order to normalize the qPCR values, each qPCR reaction was performed in three independent runs using SYBR Premix Ex Taq (Takara, Dalian, China) in Bio-Rad CFX Connect™ Real-Time PCR Detection System, with a protocol of initial denaturation at 95 °C for 3 min, followed by 40 cycles at 95 °C for 10 s, 60 °C for 20 s and 72 °C for 30 s.

To monitor the CLas persistence in *T. radiata*, newly emerged female adults of *T. radiata* (considered $F_1$ generation) were collected from the above experiment and fed with 20% honey water. After 1, 5, 10, and 15 days, 10 parasitoids were recaptured, subsequently ground for DNA extraction and CLas titer detection and quantification using qPCR. The protocol of DNA extraction and qPCR reaction was the same as above, and qPCR quantification was repeated three times for each treatment.

## Localization patterns of CLas in *Tamarixia radiata*
*Localization patterns of CLas in different instars of T. radiata.* Fluorescent in situ hybridization (FISH) was used to visualize the distribution of CLas in *T. radiata* exposed to CLas positive ACP, following the method of Gottlieb et al.[32] with a slight modification. Eggs and different larval instars of *T. radiata* were collected and fixed in Carnoy's solution (chloroform-ethanol-glacial acetic acid [6:3:1,vol/

vol] formamide) overnight at 4 °C. After fixation, the samples were washed three times in 50% ethanol with 1× phosphate buffered saline (PBS) for 5 min. Then the samples were decolorized in 6% $H_2O_2$ in ethanol for 12 h, after which they were hybridized overnight in 1 ml hybridization buffer (20 mM Tris-HCl pH 8.0, 0.9 M NaCl, 0.01% sodium dodecyl sulfate, 30% formamide) containing 10 pmol of fluorescent probes/ml in a 37 °C water bath under dark conditions. The CLas probe used for FISH was 5′-Cy3-GCCTCGCGACTTCGCAACCCAT-3′. Finally, the stained *T. radiata* samples were washed three times in a washing buffer (0.3 M NaCl, 0.03 M sodium citrate, 0.01% sodium dodecyl sulfate, 10 min per time). After the samples were whole mounted and stained, the slides were observed and photographed using a Nikon eclipse Ti-U inverted microscope. For each stage sample, approximately 20 individuals were examined to confirm the results.

*Localization patterns of CLas in different organs of T. radiata.* Different organs (gut, fat body, ovary, poison sac, salivary glands, spermatheca, and chest muscle) were dissected from newly emerged adults of *T. radiata* in 1× phosphate buffered saline (PBS) under a stereomicroscope using a depression microscope slide and a fine anatomical needle. After a sufficient number of each tissue sample was collected (20 or more), the tissues were washed three times with 1 × PBS, followed by the fixation, decolorization, and hybridization procedures as outlined above, except that this time of decolorization was 2 h. After hybridization, nuclei in the different organs were counterstained with DAPI (0.1 mg/ml in 1 × PBS) for 10 min, then the samples were transferred to slides, mounted whole in hybridization buffer, and viewed using confocal microscopy (Nikon, Japan).

## Maternal transmission of CLas between *Tamarixia* generations.
Five groups of experiments were used to clarify whether CLas can be transmitted vertically between different *T. radiata* generations. In the first group, 60 pairs of newly emerged *T. radiata* adults from the CLas-infected ACP colony (potential CLas-acquired parasitoid adults, $F_0$ generation) were introduced into 60 nylon bags (one female per cage). Each bag covered one fresh citrus plant shoot with one marked CLas-free 4th instar nymph of ACP, the parasitoid females were given 24 h to oviposit, then transferred to another four groups successively to oviposit with intervals of 24 h before they were recaptured for CLas-PCR detection (58/60 and 56/60 *T. radiata* females and males respectively were CLas-infected). Only the progeny ($F_1$ generation) in which parasitoid parents were both CLas-infected continued to be investigated.

When the $F_1$ progeny of CLas-infected parasitoid females developed to egg, larval, pupal, and adult stages respectively, they were collected and divided into two groups; in one group samples were used for the qPCR detection of the CLas titer, and the other group was used for the FISH visualization of CLas. The qPCR and FISH analysis protocols of CLas as well as the number of tested individuals were the same as previously outlined. Each stage was repeated three times.

## CLas detection in *T. radiata*-inoculated ACP
*Quantitative PCR detection of CLas.* Approximately 60 newly emerged parasitoid adult females from CLas-infected ACP hosts (potential CLas-acquired parasitoid adults) were collected using an aspirator. They were first starved for 5 h, then released into finger tubes (diameter 6 mm × length 30 mm); one female per tube containing one 4th instar nymph of CLas-free ACP (this was treated as one experimental replicate). The probing behavior of the parasitoids was observed under a stereomicroscope, after which the parasitoids were recaptured for CLas PCR detection (similar to the above experiment, approximately 95% were CLas-infected). Only those 4th instar ACP nymphs, probed for egg-laying by a CLas-infected parasitoid but survived from the probing (the averaged proportion of such samples was 5.36 ± 0.47% and were 100% CLas infected), were transferred onto fresh CLas-free M. exotica shoots to complete their development (hereafter referred as "*T. radiata*-inoculated ACP"). The experiment was repeated in 32 parallel replicates (Supplementary Table 1), in which 103 *T. radiata*-inoculated ACP nymphs were finally obtained.

Following the above, thirty *T. radiata*-inoculated ACP nymphs were collected when they developed into 5th instar nymphs (the stage when infection proliferation might have just begun since the infection was introduced at the 4th instar). In addition, thirty 8-day old adults that developed from the *T. radiata*-inoculated ACP nymphs were also collected. This was because the results in Wu et al.[28] revealed that the proportion of CLas-infected ACP individuals exceeds 90% at the 12th day after infection acquisition, while ACP takes 4 days to develop into an adult from 5th instar stage. Their alimentary canals and salivary glands were dissected under a stereomicroscope using the methods of Ammar et al.[33], and hemolymphs were collected with a 10 μl pipette tip using the method of Killiny et al.[34]. The DNA of the alimentary canals, salivary glands and hemolymphs were extracted using TIANamp Micro DNA Kit (Tiangen, Beijing, China), and the relative titers of CLas in each tissue of ACP nymphs and adults were detected by qPCR with of LJ900. The *β-actin* gene of ACP (F 5′-CCCTGGACTTTGAACA GGAA-3′; R 5′-CTCGTGGATACCGCAAGATT-3′) was selected as an internal control for data normalization and quantification[35]. For each sample, qPCR detection was repeated three times.

*FISH visualization of CLas.* The alimentary canals and salivary glands of 5th instar nymphs and 8-day old adults of *T. radiata*-inoculated ACP were dissected as described above, and the distribution of CLas was visualized by FISH and confocal

microscopy. The alimentary canals and salivary glands of CLas-infected ACP nymphs and adults (collected from CLas-infected citrus plants) were used as a positive control, and five to ten samples were detected by FISH for each tissue.

*CLas transmission from* T. radiata*-inoculated ACP to citrus plants*. According to the above experimental results, if the CLas could be detected in the salivary glands of the 8-day old ACP adults (T. radiata-inoculated ACP), 30 more of these adults were randomly selected to inoculate on fresh shoots of CLas-free citrus. ACP adults that acquired CLas from plants and CLas-free ACP adults were used as positive and negative controls respectively.

After 20, 30, 40, and 50 days of feeding samples of the citrus leaves fed on by T. radiata-inoculated ACP (named as CLas-recipient citrus leaves), by ACP that acquired CLas from plants (positive control), and fed on by CLas-free ACP (negative control) were cut (1 cm2). Their DNAs were extracted using DNAsecure Plant Kit (Tiangen, Beijing, China). The infections of CLas in these plants were detected by nested PCR based on the methods of Jagoueix et al.[36] and Deng et al.[37]. The experiment was repeated in six plants for each of 20, 30, 40, and 50 days feeding duration, and the infection rates of CLas were calculated.

**Localization of CLas in citrus plants fed on by *T. radiata*-inoculated ACP**. In order to further confirm the infection of CLas in the recipient citrus leaves, FISH was used to visualize the localization of CLas. According to the above experimental results, after being fed on for 50 days by the T. radiata-inoculated ACP adults, citrus leaf sections containing the midrib were cross-sliced in 30 μ sections using a cryostat (CM1950, Leica, Germany). The leaf samples were prepared for FISH vitalization according to the protocol described by Gottlieb et al.[32]. Citrus leaves from the plant that had been fed on by ACP adults that acquired CLas from plants and CLas-free ACP adults were used as positive and negative controls, respectively. Five to 10 leaf samples were detected by FISH for each treatment.

**Phylogenetic analysis of CLas bacteria in different ACP populations and citrus plants**. To assess the identity of the CLas bacteria in CLas donor ACP, CLas vectored parasitoids, T. radiata-inoculated ACP and recipient citrus leaves, the outer membrane protein gene (*omp*) of CLas was PCR amplified with the primers HP1asinv (5′-GATGATAGG TGCATAAAAGTACAGAAG-3′) and Lp1c (5′-AATACCCTTATGGGATACAAAAA-3′) following the procedure described in Bastianel et al.[38]. Then the PCR products were sent for sequencing after visualizing the expected bands on 1% agarose gels.

All the DNA sequences of CLas *omp* gene were edited and aligned manually using Clustal X1.83[39] in Mega 6[40]. The best model and partitioning scheme were chosen using the Bayesian information criterion in PartitionFinder v.1.0.1[41]. Phylogenetic analysis was undertaken using a maximum likelihood (ML) method with 1000 non-parametric bootstrap replications in RAxML[42]. *Escherichia coli* was used as an outgroup.

**Statistics and reproducibility**. Taking *18S rRNA* gene of T. radiata and the β-actin gene of ACP as housekeeping genes, the relative titers of CLas in different stages and different tissues of T. radiata and ACP were calculated using the method of $2^{[-\Delta\Delta ct}$ [43]. For the parallel experiments that had more than three replicates the differences were compared using analysis of variance (ANOVA) with SPSS 18.0 at a significance level α = 0.05; while for CLas titer, two-sample comparison between genders of *Tamarixia* adults analysis was performed using paired t-test. Fluorescent pictures were processed using Photoshop CS5 software.

**Reporting summary**. Further information on research design is available in the Nature Research Reporting Summary linked to this article.

## Data availability

The *omp* gene sequences of CLas in the donor ACP, *Tamarixia radiata* parasitoids, T. radiata-inoculated ACP, and recipient citrus plants are available in GenBank (accession numbers: MT191373–MT191376); Correspondence and requests for materials should be addressed to Bao-Li Qiu.

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

## Acknowledgements

The authors thank anonymous reviewers for their valuable comments and suggestions on the earlier version of this manuscript. This work was supported by the Guangdong Laboratory of Lingnan Modern Agriculture Project (NT2021003), the NSFC-Guangdong Joint Research Fund (U1701231), the National High Level Talent Special Support Plan (2020), the Key-Area Research and Development Program of Guangdong Province (2018B020205003) to B.L.Q. The funders had no role in the study design, data collection, and interpretation, or the decision to submit the work for publication.

## Author contributions

B.L.Q., C.F.G. and M.Z.A. designed the study; C.F.G., D.O. and L.H.Z. carried out the experiments; C.F.G., M.Z.A., D.O. and L.H.Z. participated in data analysis and carried out sequence alignments; Z.T.L. and W.S. carried out the photographs and analyses; B.L.Q., M.Z.A., R.G.S. and C.L.M. wrote the manuscript. All authors gave final approval for publication.

## Competing interests

The authors declare no competing interests.
