## [Peer Review File · Communications Biology]

Reviewers' comments:

Reviewer #1 (Remarks to the Author):

Additional comments are on the annotated pdf.

Review of COMMSBIO-20-3159-T

This paper reports an interesting and important finding, that the primary parasitoid of ACP can transmit the HLB pathogen horizontally in the process of probing hosts. The authors have performed a battery of experiments to demonstrate, using the molecular techniques of PCR and FISH, that the parasitoid is capable of transmitting the pathogen, that ACP nymphs so exposed become infective to citrus plants, and that parasitoids developing in infected hosts themselves become infected and thus potential vectors to non-infected hosts, although transgenerational transmission vertically through eggs does not occur. They also localized the pathogen in various tissues of the parasitoid throughout its various stages of development.

Unfortunately, the presentation of the paper is very sloppy and the English composition is abysmal, with poor sentence structure, extraneous verbiage (see my various deletions in the annotated pdf), and broken English in many places. It made for a real mess to read. The first-language English authors really need to spend more time improving the quality of the writing before any resubmission. By way of example, I have done some editing to the Abstract to improve readability, but the whole paper needs revision for English composition. It is excessive to expect a reviewer to do this. I would also caution that the authors have not shown that the potential role of the parasitoid as a disease vector is sufficient to fully counter or negate its value as a BC agent, thus my advice to change the title to "A parasitoid vectors a plant disease, *potentially* diminishing the benefits it confers as a BC agent."

Line 48. This last sentence is a mess. How about this: "We illustrate, for the first time, the formerly unrecognized risk that a parasitoid can potentially serve as a phoretic vector of the pathogen transmitted by its host, thus potentially diminishing some of the benefits it confers via biological control."

Line 65. Don't you mean "resulted in the evolution of insecticide resistance..."?

Line 78. An example of careless sentence structure. ACP feeding alone is not enough to 'spread' CLas - it requires a sequence of events, beginning with acquisition and ending in transmission to a susceptible host plant.

Line 80. Avoid the use of gratuitous or superlative adjectives and adverbs in scientific writing. Delete 'incessantly'. "between different" is redundant.

Line 81. "management efficiency of CLas spread" - this reads like an oxymoron. You probably mean 'diminish the benefits of ACP biological control via horizontal transmission of the pathogen'.

Line 85. Final sentence is another English abomination with singular/plural lack of agreement and other problems.

Line 95. Either "in males" and "in females" or "of males" and "of females".

Line 96. Incorrect use of article 'the'.

Line 97. "levels" (plural). Adults 'emerge' - only eggs 'eclose'.

Line 100. "spermary" is not a word.

Line 101. There are many muscles - which ones? If all, say 'muscle tissues'.

Line 103. Compound subjects are rarely advisable, and 'infection' and 'localization' do not belong together. "CLas infections in *T. radiata* were localized visually using FISH..."

Line 106. "visually confirmed" - not "positively visualized", here and elsewhere. "...in all larval stages..., but not in the eggs..." (eggs are not larval stages).

Line 108. When is development not continuous? You mean "over the course of development..."

Line 116. Maternal transmission is, by definition, transgenerational.

Line 123. Do not use the comparative without completing the comparison: stronger than what? This sentence contains no useful information whatsoever.

Line 125. "...until it almost disappeared..."

Line 128. Another awkward heading. "Localization of CLas in ACP following inoculation via parasitoid probes."

Line 130. Avoid pointless verbiage such as "Results revealed that..." You are just wasting space. And be more precise. "ACP nymphs probed by a CLas-infected parasitoid..." Probed is more exact

than "exposed to'.

Line 133. They don't develop to 8-day old adults - development is complete at adult emergence. Your writing is so sloppy...

Line 136. Another clumsy sentence that needs rephrasing.

Line 151. I suggest using "T. radiata-inoculated ACP" instead of "CLas-recipient ACP."

Line 154. Combine this and the following sentences into one. "After 40 d of feeding by T. radiata-inoculated ACP, approximately one third of citrus leaves were CLas positive, compared to 100% of leaves fed on by ACP that acquired CLas from plants, but after 50 d, 100% of the former also tested positive. These results demonstrated that CLas can proliferate within citrus plants following infection by ACP adults inoculated as nymphs by T. radiata probing."

Line 165. Sentence makes no logical statement.

The Discussion should begin by making logical inferences from the results presented. The whole first paragraph is devoted to literature review. It is fine to cite relevant and supporting work, but do it in the context of discussing direct inferences from your own results. As such, the second paragraph is a much better starting point for the discussion. Eliminate the first paragraph, but cite the relevant studies in support of your own findings.

Line 179. "Parasitoids AND mites..."

Line 192. "...a parasitoid..."

Line 199. at least 5 days after what?

Line 207. "spread' by what? arthropod vectors, one must assume.

Line 231. "...there is a chance of CLas introduction." Introduction is, by definition, and initiation of infections. Look more closely at your word choices.

Reviewer #2 (Remarks to the Author):

These authors propose a mechanism by which parasitoids pass a plant pathogen (the citrus greening bacterium, CLas) from infected to uninfected CLas vectors (Asian citrus psyllid) in a way that allows the pathogen to be transmitted to plants. Since the psyllids transmit the pathogens to trees as adults and parasitoids (by definition) kill their hosts (in this case in the nymphal stage) this would seem to be an impossible pathway, but the authors argue that it is possible since some hosts survive the stinging process, which is when the hosts become inoculated. The results presented elegantly show some of the steps that would be necessary for this pathway to occur, but I am unconvinced that *all* of the steps have been experimentally demonstrated. The main problem is in the experiment on transmission of CLas from infected to uninfected hosts via the parasitoids. According to the authors, 'approximately 60 parasitoids' were used for this experiment, each of which was placed within a single 'finger vial' with a single host nymph. As expected, most of these hosts were stung and died (and thus could not take part in the pathway), but the authors note that 5% survived and were used for subsequent assays. 5% of 60 = 3, which means that there should be about 3 nymphs available for those assays. We are given no numbers for how many of the 'approximately 60' parasitoids were infected or how many of the surviving nymphs were infected, but it is clear from the results that more than 30 nymphs were used in assays testing these nymphs. I have read this section over many times and have to conclude that the numbers are either impossible or the methods are incorrectly explained. Since I can only judge what is in front of me, I must conclude that there was something wrong with the experiment and so cannot accept the results. And without these results, the entire argument is flawed. I explain this criticism in more detail below.

In general, while the paper is impressive in its methods and some of the results, there are many gaps in reporting. In particular we are never given sample sizes and so can not have any idea of things like the percentage of parasitoids that become infected or the percentage of infected parasitoids that are capable of infecting hosts. While these quantities are not needed to know whether or not transmission *can* occur, they are crucial to understanding the risk of them occurring.

The writing in the paper is impressive for non-native English speakers and is mostly adequate for understanding what was done but there are numerous problems with grammar; I did not have

time to correct these.

Abstract:

L37. 'could' is too vague – it could be said about almost anything. If you can't say 'can' then the work is not worth publishing.

L39. 'between'? seems wrong

L43. Not clear what 'check' refers to

Introduction:

L57. Please don't talk about 'nightmares' in a scientific article.

L65. 'a long time' is too vague

First two paragraphs of introduction: Please give some indication where the disease is native and where it was introduced. This is important if you are going to talk about biological control introductions (by the way the term 'classical biological control' is jargon – a better term is 'importation biological control'). I believe that the pathogen is native to China and so it is confusing to hear about effects of *Tamarixia* in China as a result of 'classical' biological control (L75-76).

L73 – 77. The authors state that biological control releases have led to reduction in ACP and then seem to illustrate this by providing information on parasitism rate. However, parasitism rate does not tell us about ACP decline. It is certainly consistent with decline but it does not always work that way.

Methods

L255 A lot of detail is given on the CLas-free plants but none for the CLas-infected plants. Please add some details (e.g. where are they from).

L266. I think you mean 'removed' not 'cleaned off'

Part 3 (L261). A subset of the parasitoids should have been tested to ensure that they were CLas-free as part of this experiment. From this protocol, all we know is whether parasitoids tested positive for CLas after exposure. I know it seems unlikely, but given the protocol, is possible that the parasitoids were infected with CLas before the experiment. I'm not sure how crucial this is to the conclusions, but it is a flaw in the design; the experiment is essentially missing a negative control.

L311. I think you mean 'spermatheca' instead of 'sprermary'

L321. Delete 'further'

L341. Why were the parasitoids starved for 5 hours?

L342. I presume these were ACP that were *free* of CLas; this has to be made clear or the experiment makes no sense.

L344-347. I suppose it is obvious that the only way that the transmission pathway envisioned by the authors could take place is if ACP nymphs survive stinging encounters with the parasitoids, an event that is expected to be very rare. In this experiment, the authors report that this occurred in 5.4% of exposures (this value is called an 'averaged proportion' [L346] although we are not told over what kind of groups such an average could be taken, so this needs to be cleared up).

However, it is important to note that the exposure method was highly artificial, with nymphs apparently taken off of their host plants and placed into glass 'finger tubes' for exposure to parasitoids (actually the method is not fully described at all – but this is what I gather). It would be important to know whether this exposure method altered the proportion of nymphs surviving parasitoid sting events. One possibility is that this exposure made it more difficult for parasitoids to properly handle their hosts and thus that the host survival rate is artificially high (I suppose it could also be artificially low). This is a very important point since the entire transmission scenario is based on the 5% of nymphs that survive parasitoid stinging if I understand it correctly.

L348. Please say *how many* ACP nymphs (and adults) were used for the assays described in this paragraph. Since there were 'approximately 60' parasitoids and each parasitoid was given a sing

host nymph, and only 5% of these survived (from the last paragraph), there should be only about 3 nymphs available for this set of assays. Same comment for the insects used for FISH in the paragraph starting on L362.

L369. Based on the description of the methods interpreted above, it would be *impossible* to have 30 ACP adults with which to do the plant inoculation experiments. I don't know how I will be able to interpret these results given this consideration. I think I will have to consider them invalid.

L403-4. What is meant by 'relative titers'? Relative to what? The beta-actin gene? Please be more clear about this. I would like to see some more detail about this 'method 2(-delta delta)'. I realize that a reference is given and I presume that the method is correct, but just a summary of the methods would be helpful.

L405. The way I understand it, ANOVA is done on a replication level of 3, which is unusually low. I think it may be fine but it would be nice for readers to be assured that the assumptions of ANOVA were met by the data.

Results

Fig. 1. I'm not quite sure what is meant by 'relative titer . . .' Anyway, the finding of zero in eggs is good; it somewhat makes up for the error of not having a negative control for the parasitoids (noted above). You should be sure to mention this in the text (L90 - 97); I suggest writing something like, 'as expected, levels were zero in eggs . . .'

Fig. 2 legend. I believe you mean 'ovipositor' and not 'poison needle' (also to match the text).

L108-109. I like the pictures in Fig. 3, but I think it's a bit overstating the results to say that they show spread from the mouthparts of the abdomen. This statement seems to come from one photo in which it's not clear what side of the larva has the mouth vs. the anus. If this statement is based upon more data please indicate that.

More importantly - we are not told (i) how many individuals were investigated for these analyses, or (ii) of those, how many showed positive results, or (iii) if the description of the FISH results are from more than one individual, how consistent the results were from individual to individual. I have the same comment for all of the FISH results.

L115. Why is the DuFour's gland mentioned here for the first time? It was not mentioned in the methods. If you really did find this gland, please explain how you identified it as such.

Section starting on L116. Please recall that we were told in the methods (L327-8) that only parasitoids testing positive for CLAs would be used for these analyses. We need to be told what the proportion positive was (you can provide that information in the methods section or here, but you need to provide it).

L117-120. This sentence is too long and difficult to understand. I had to read it multiple times to understand it. Please work on clarifying this. For instance, the titer was higher in eggs than what? I would suggest breaking this sentence up into 2 or 3.

Fig. 6. I'm not sure how the results in Fig. 6 can be compared to the results in Fig. 1. Both say 'relative titers', but only in Fig. 6 are we told about the fold change. It would be great to have all of the figures showing the qPCR results displayed in a way that they can be easily compared to one another (maybe that's how this was done but if so this needs to be made more clear).

Section starting on L128. In this section we need to be told the percentage of surviving nymphs that were positive for CLAs. Also - the proportion of parasitoids testing positive (from L344).

L136. Please don't call the parasitoid a 'phoretic vector' at this point. This is the hypothesis you are testing; you are treating it like a conclusion.

Section starting on L151. As for all of the other results, there is no indication of sample size here.

Discussion

Based upon my skepticism of the results (outlined above) I disagree with parts of the Discussion section. I do agree with the recommendation that CLas-free parasitoids be used in releases since I do believe that you have shown the potential for vectoring under the special circumstance of hosts surviving an attack.

Reviewer #3 (Remarks to the Author):

In their manuscript, the team of Prof. Qiu present a new route for plant pathogen spread. Parasitoids that interacted with herbivore hosts that vector plant pathogens, became carriers of the plant pathogen. By visiting herbivores on other trees, the parasitoid may vector the plant disease via parasitizing or probing herbivores feeding on this tree. Overall, I find that the authors provide convincing evidence of pathogen being vectored by parasitoids, with its most critical experiment confirming pathogen infection in plants after herbivores that interacted with parasitoids fed from these plants. I have no technical reservations, the study seems to be carefully conducted. I do have a number of suggestions for this manuscript to connect to a broader readership:

1) The introduction focusses on a single plant-pathogen-herbivore-parasitoid system. The broader context of disease spread by insects and how these interactions may be affected by parasitoids could make the paper more interesting to readers that are not working on citrus pests.

2) Following this suggestion, I miss a more general picture of what is already known from the role of parasitoids in herbivore vectored plant diseases. Several studies have evaluated how parasitoids as biocontrol agents may affect disease spread and biocontrol may not yield the desired effect (or even promote disease spread). Parasitoids/predators may affect herbivore behaviour and cause increase in disease spread by increasing herbivore movement among plants. Plant pathogens may affect recruitment of parasitoids. Herbivores infected with plant diseases may affect performance of parasitoids. To me the results of this study still add a novel component to insect vectored diseases, but when placing those results in the context I suggest here would increase the value of the findings. The connection to broader literature on the role of parasitoids in plant disease spread could be discussed in both the introduction and discussion.

By line number:

Line 35: The rationale of using parasitoids... Has never been evaluated. This is not entirely true in the context that I sketch in my general feedback. It is known that parasitoids may have unexpected negative effects by increasing plant disease spread. It is not known that parasitoids themselves may vector the disease (line 36).

Line 36: Change to "...we report that the most dominant parasitoid of ACP (*Tamarixia radiata*) could vector.."

Line 41: This sentence is unclear

Line 48: this sentence is long and complex. I also suggest to delete the "for the first time" here. This is already stated in line 36.

Line 91 and elsewhere: The results section could be condensed by using a more direct writing style. Remove "Results showed that the titers..." and start right away with "The titers..."

Line 94: Here you highlight that parasitoid males have high titers of the plant pathogen. This aspect is not discussed and the consequences for disease spread could be added to the discussion (or I missed this?).

Line 98: add s to adult

Line 111-115: this paragraph describes similar results as the previous, but using a different technique. I suggest to integrate the two paragraphs

Line 122: change transmitted to transmit

Line 182: entosymbiont should be endosymbiont

Line 266: how were the individual marked?

We have revised the manuscript in detail according to the valuable comments and suggestion from reviewers. All authors have read and agree with this revised version. Please find our response and revision in the following.

Reviewer #1 (Remarks to the Author):

Additional comments are on the annotated pdf.

Review of COMMSBIO-20-3159-T

This paper reports an interesting and important finding, that the primary parasitoid of ACP can transmit the HLB pathogen horizontally in the process of probing hosts. The authors have performed a battery of experiments to demonstrate, using the molecular techniques of PCR and FISH, that the parasitoid is capable of transmitting the pathogen, that ACP nymphs so exposed become infective to citrus plants, and that parasitoids developing in infected hosts themselves become infected and thus potential vectors to non-infected hosts, although transgenerational transmission vertically through eggs does not occur. They also localized the pathogen in various tissues of the parasitoid throughout its various stages of development.

Unfortunately, the presentation of the paper is very sloppy and the English composition is abysmal, with poor sentence structure, extraneous verbiage (see my various deletions in the annotated pdf), and broken English in many places. It made for a real mess to read. The first-language English authors really need to spend more time improving the quality of the writing before any resubmission.

*By way of example, I have done some editing to the Abstract to improve readability, but the whole paper needs revision for English composition. It is excessive to expect a reviewer to do this. I would also caution that the authors have not shown that the potential role of the parasitoid as a disease vector is sufficient to fully counter or negate its value as a BC agent, thus my advice to change the title to "A parasitoid vectors a plant disease, *potentially* diminishing the benefits it confers as a BC agent."*

We thank the reviewer for the comment. We apologize for mistakes in the language of the manuscript. To rectify this, we have sought additional help from the native English-speaking co-authors and a colleague, Dr. Andrew Cuthbertson (York, UK). Dr. Cuthbertson has been a collaborator on several of our past projects and he himself has published over 130 peer-reviewed papers. He now assures us that the English language in the manuscript is adequate.

We have also accepted the suggestion for the title. The manuscript title now reads "Parasitoid vectors a plant pathogen, potentially diminishing the benefits it confers as a biological control agent".

Specific Points:

1) Line 48. This last sentence is a mess. How about this: "We illustrate, for the first time, the formerly unrecognized risk that a parasitoid can potentially serve as a phoretic vector of the pathogen transmitted by its host, thus potentially diminishing some of the benefits it confers

via biological control."

We thank reviewer for this suggestion. We have rewrote this sentence according to the suggestion with minor reduction due to the word limitation in Abstract line 37-39 (in current revised new version, same as below).

2) *Line 65. Don't you mean "resulted in the evolution of insecticide resistance...?"*

We thank the reviewer for pointing this out. We have reworded the sentence in line 69.

3) *Line 78. An example of careless sentence structure. ACP feeding alone is not enough to 'spread' Clas - it requires a sequence of events, beginning with acquisition and ending in transmission to a susceptible host plant.*

We have reworded the sentence in question. It now reads: "Prior studies have revealed two main routes for the increasing distribution of the CLas pathogen; grafting and ACP feeding between CLas infected and healthy but susceptible citrus plants" in line 84-86.

4) *Line 80. Avoid the use of gratuitous or superlative adjectives and adverbs in scientific writing. Delete 'incessantly'. "between different" is redundant.*

We thank the reviewer for this comment. We have removed the word as suggested, in line 86-87.

5) *Line 81. "management efficiency of Clas spread" - this reads like an oxymoron. You probably mean 'diminish the benefits of ACP biological control via horizontal transmission of the pathogen'.*

We do apologize for this poor use of language. We thank the reviewer for pointing it out and suggesting an alternative wording. We have fixed the sentence here in the Introduction line 88-89 as well as in the Discussion sections.

6) *Line 85. Final sentence is another English abomination with singular/plural lack of agreement and other problems.*

Again, we apologize for this. We have corrected the sentence in line 98-101.

7) *Line 95. Either "in males" and "in females" or "of males" and "of females".*

We have corrected as suggested. We have made it "of males" and "of females" in line 107-108.

8) *Line 96. Incorrect use of article 'the'.*

We have corrected the sentence in question in line 109.

9) *Line 97. "levels" (plural). Adults 'emerge' - only eggs 'eclose'.*

We have corrected the words in is sentence as pointed out by the reviewer. It now reads: '.... undetectable **levels** and totally disappeared from qPCR detection on the 10th day after **emergence** (Figure 1C) in line 110.'

10) *Line 100. "spermary" is not a word.*

We apologize for this. The word has been changed to “spermatheca” in line 113.

11) *Line 101. There are many muscles - which ones? If all, say 'muscle tissues'.*

We thank the reviewer for this comment. We have changed it to chest muscles in line 114.

12) *Line 103. Compound subjects are rarely advisable, and " and 'localization' do not belong together. "CLas infections in T. radiata were localized visually using FISH..."*

We have corrected the sentence as suggested in line 117.

13) *Line 106. "visually confirmed" - not "positively visualized", here and elsewhere. "...in all larval stages..., but not in the eggs..." (eggs are not larval stages).*

We have corrected the sentence as suggested by the reviewer in line 118.

14) *Line 108. When is development not continuous? You mean "over the course of development..."*

We have corrected the sentence as suggested by the Reviewer in line 120-121.

15) *Line 116. Maternal transmission is, by definition, transgenerational.*

We agree with the reviewer. However, here we want to clearly define it between *Tamarixia* parasitoid generations, not ACP generations, line 132.

16) *Line 123. Do not use the comparative without completing the comparison: stronger than what? This sentence contains no useful information whatsoever.*

We have replaced the word “stronger” with “strong” in line 138. Thank you.

17) *Line 125. "...until it almost disappeared..."*

We have corrected the sentence in line 140.

18) *Line 128. Another awkward heading. "Localization of CLas in ACP following inoculation via parasitoid probes."*

We have revised the heading as suggested by the reviewer in line 143.

19) *Line 130. Avoid pointless verbiage such as "Results revealed that..." You are just wasting space. And be more precise. "ACP nymphs probed by a CLas-infected parasitoid..." Probed is more exact than "exposed to'.*

We thank the reviewer for this advice. We have revised the sentence as suggested in line 144-146.

20) *Line 133. They don't develop to 8-day old adults - development is complete at adult emergence. Your writing is so sloppy...*

We apologize for this. We have revised the sentence in question in line 147.

21) *Line 136. Another clumsy sentence that needs rephrasing.*

Again, we apologize for this. We have revised the sentence in question. It now reads: “All

these findings suggest that after inoculation by *T. radiata*, CLas titer accumulates in the hemolymph of 5th instar ACP nymphs as well as in the hemolymph and salivary glands of 8-day old ACP adults, but fails to enter their midguts (Figure 6)" in line 149-152.

22) Line 151. I suggest using "*T. radiata*-inoculated ACP" instead of "CLas-recipient ACP. We have replaced the term throughout the MS as suggested in line 163 and all the other places in the whole text.

23) Line 154. Combine this and the following sentences into one. "*After 40 d of feeding by T. radiata*-inoculated ACP, approximately one third of citrus leaves were CLas positive, compared to 100% of leaves fed on by ACP that acquired CLas from plants, but after 50 d, 100% of the former also tested positive. These results demonstrated that CLas can proliferate within citrus plants following infection by ACP adults inoculated as nymphs by *T. radiata* probing."

We thank the reviewer for this suggestion. We have revised the sentences accordingly in line 166-170.

24) Line 165. Sentence makes no logical statement. We have adjusted it line 173.

25) *The Discussion should begin by making logical inferences from the results presented. The whole first paragraph is devoted to literature review. It is fine to cite relevant and supporting work, but do it in the context of discussing direct inferences from your own results. As such, the second paragraph is a much better starting point for the discussion. Eliminate the first paragraph, but cite the relevant studies in support of your own findings.*

We thank the reviewer for their suggestions. We have changed the order of first and second paragraphs in the text and sought to take the advice in the Discussion section.

26) Line 179. "*Parasitoids AND mites...*" We have inserted the word 'and' as suggested line 195.

27) Line 192. "*...a parasitoid...*" We have corrected the sentence in line 209.

28) Line 199. *at least 5 days after what?* We apologize for this mistake. We have completed the sentence. It now reads: '... and that it can persist in the parasitoid adult for at least 5 days following wasp emergence' in line 254.

29) Line 207. "*spread' by what? arthropod vectors, one must assume.*" We have completed the sentence in question. It now reads: 'To our knowledge, this is the first report of an insect natural enemy, which is actively applied for the biological control of a given pest, that has been found to contribute to transmission of a plant pathogen and therefore spread of the resulting plant disease' in line 258-261.

30) Line 231. "...there is a chance of CLas introduction." Introduction is, by definition, and initiation of infections. Look more closely at your word choices.

We have deleted the words 'Introduction and' in line 248-249.

We have also revised the manuscript according to the comments made directly to the PDF as suggested by the reviewer.

Reviewer #2 (Remarks to the Author):

*These authors propose a mechanism by which parasitoids pass a plant pathogen (the citrus greening bacterium, CLas) from infected to uninfected CLas vectors (Asian citrus psyllid) in a way that allows the pathogen to be transmitted to plants. Since the psyllids transmit the pathogens to trees as adults and parasitoids (by definition) kill their hosts (in this case in the nymphal stage) this would seem to be an impossible pathway, but the authors argue that it is possible since some hosts survive the stinging process, which is when the hosts become inoculated. The results presented elegantly show some of the steps that would be necessary for this pathway to occur; but I am unconvinced that *all* of the steps have been experimentally demonstrated. The main problem is in the experiment on transmission of CLas from infected to uninfected hosts via the parasitoids. According to the authors, 'approximately 60 parasitoids' were used for this experiment, each of which was placed within a single 'finger vial' with a single host nymph. As expected, most of these hosts were stung and died (and thus could not take part in the pathway), but the authors note that 5% survived and were used for subsequent assays. $5\% \text{ of } 60 = 3$, which means that there should be about 3 nymphs available for those assays. We are given no numbers for how many of the 'approximately 60' parasitoids were infected or how many of the surviving nymphs were infected, but it is clear from the results that more than 30 nymphs were used in assays testing these nymphs. I have read this section over many times and have to conclude that the numbers are either impossible or the methods are incorrectly explained. Since I can only judge what is in front of me, I must conclude that there was something wrong with the experiment and so cannot accept the results. And without these results, the entire argument is flawed. I explain this criticism in more detail below.*

*In general, while the paper is impressive in its methods and some of the results, there are many gaps in reporting. In particular we are never given sample sizes and so can not have any idea of things like the percentage of parasitoids that become infected or the percentage of infected parasitoids that are capable of infecting hosts. While these quantities are not needed to know whether or not transmission *can* occur; they are crucial to understanding the risk of them occurring.*

We are very grateful to the reviewer for the constructive comments received. We have sought to revise the manuscript in accordance to the comments, especially concerning the numbers of *Tamarixia*-inoculated ACP nymphs. Please find the detail response below to query numbers 16 & 17.

The writing in the paper is impressive for non-native English speakers and is mostly adequate for understanding what was done but there are numerous problems with grammar; I did not have time to correct these.

We thank the reviewer for this supportive comment. We do apologize for mistakes in the language of the manuscript. To rectify this, we have sought additional help from a native English-speaking co-authors and a colleague, Dr. Andrew Cuthbertson (York, UK). Dr. Cuthbertson has been a collaborator on several of our past projects and he himself has published approximately 130 peer-reviewed papers. He now assures us that the English language in the manuscript is adequate.

Specific points:

Abstract:

1) L37. *'could' is too vague – it could be said about almost anything. If you can't say 'can' then the work is not worth publishing.*

We have changed the word to “can” in line 34. Our original intention was to say the wasp has the possibility to vector HLB pathogen.

2) L39. *'between' seems wrong*

We have deleted this word, line 35.

3) L43. *Not clear what 'check' refers to*

We have revised the sentence with the word “probing” in line 36, i.e. the behavior of parasitization.

Introduction:

4) L57. *Please don't talk about 'nightmares' in a scientific article.*

We have removed it.

5) L65. *'a long time' is too vague*

We have revised the sentence in question to now read: ‘Chemical control has been used aggressively to eliminate ACP from citrus orchards in many countries. However, frequent use of insecticides has resulted in the evolution of insecticide resistance for ACP in many parts of the world....’ in line 68-70.

6) *First two paragraphs of introduction: Please give some indication where the disease is native and where it was introduced.*

We thank the reviewer for this suggestion. We have mentioned now the origin of HLB in the first paragraph in line 60-61.

7) *This is important if you are going to talk about biological control introductions (by the way the term 'classical biological control' is jargon – a better term is 'importation biological control').*

I believe that the pathogen is native to China and so it is confusing to hear about effects of

Tamarixia in China as a result of ‘classical’ biological control (L75-76).

We thank the reviewer for this information. We have deleted the word “classical” in the text, line 73.

8) L73 – 77. *The authors state that biological control releases have led to reduction in ACP and then seem to illustrate this by providing information on parasitism rate. However, parasitism rate does not tell us about ACP decline. It is certainly consistent with decline but it does not always work that way.*

We did not mean to emphasize that the release of *Tamarixia* parasitoid led to ACP population decline in those studies. There is one out those four studies here, Pluke et al., which indicated that *Tamarixia* parasitoid can cause the decline in the population of ACP in the field. The aim we cited these four references in line 75-78 is to show that, *Tamarixia* parasitoid is a dominant parasitoid species of ACP.

Methods:

9) L255 *A lot of detail is given on the CLas-free plants but none for the CLas-infected plants. Please add some details (e.g. where are they from).*

We have added the detail of the CLas-free and CLas infected plants in line 278-279.

10) L266. *I think you mean ‘removed’ not ‘cleaned off’*

Yes. We have corrected the wording in line 290.

11) Part 3 (L261). *A subset of the parasitoids should have been tested to ensure that they were CLas-free as part of this experiment. From this protocol, all we know is whether parasitoids tested positive for CLas after exposure. I know it seems unlikely, but given the protocol, is possible that the parasitoids were infected with CLas before the experiment. I’m not sure how crucial this is to the conclusions, but it is a flaw in the design; the experiment is essentially missing a negative control.*

We thank the reviewer for this point. We did test parasitoids to ensure they were CLas free. We have revised the sentence this time to make it clearer “...randomly selected from the population that has been tested to be CLas-free....”in line 291-292.

12) L311. *I think you mean ‘spermatheca’ instead of ‘sprermary’*

Thanks. We have now corrected it in line 337.

13) L321. *Delete ‘further’*

We have deleted the word as suggested and revised the paragraph, line 347.

14) L341. *Why were the parasitoids starved for 5 hours?*

The reason for this was to increase their hunger level. This would then make the parasitoids more efficient in their checking and probing behavior.

15) L342. *I presume these were ACP that were *free* of CLas; this has to be made clear or the experiment makes no sense.*

We have added this information in the text to make it clearer in line 368.

16) L344-347. *I suppose it is obvious that the only way that the transmission pathway envisioned by the authors could take place is if ACP nymphs survive stinging encounters with the parasitoids, an event that is expected to be very rare. In this experiment, the authors report that this occurred in 5.4% of exposures (this value is called an ‘averaged proportion’ [L346] although we are not told over what kind of groups such an average could be taken, so this needs to be cleared up). However, it is important to note that the exposure method was highly artificial, with nymphs apparently taken off of their host plants and placed into glass ‘finger tubes’ for exposure to parasitoids (actually the method is not fully described at all – but this is what I gather). It would be important to know whether this exposure method altered the proportion of nymphs surviving parasitoid sting events. One possibility is that this exposure made it more difficult for parasitoids to properly handle their hosts and thus that the host survival rate is artificially high (I suppose it could also be artificially low). This is a very important point since the entire transmission scenario is based on the 5% of nymphs that survive parasitoid stinging if I understand it correctly.*

We agree with the reviewer’s comments. However, please be aware that the critical novel aspect of the current study determines that the *Tamarixia* parasitoid can vector HLB pathogen, thus, diminishing its benefit as a biological control agent, not the data of 5.36%.

We can say that the experiment design is artificial, but 5.36% is only the data we recorded in the finger tube. When releasing the parasitoid into a petri dish, a cage, or an orchard, the survival rate of probed-ACP nymphs may increase to 6%, 7%, or even higher. More ACP nymphs are available in a larger space, and the parasitoids have more choice during the probing-checking-oviposition process.

17) L348. *Please say *how many* ACP nymphs (and adults) were used for the assays described in this paragraph. Since there were ‘approximately 60’ parasitoids and each parasitoid was given a single host nymph, and only 5% of these survived (from the last paragraph), there should be only about 3 nymphs available for this set of assays. Same comment for the insects used for FISH in the paragraph starting on L362.*

We have offered a supplementary table for the raw data from the experiment (Table S1) and listed the number of 5th instar nymphs and 8-d old adults of ACP we got in each experiment. To get enough probed-but-survived ACP 5th instar nymphs, we repeated the experiment using more than 30 parallel groups; we finally got approximately 100 target nymphs for further experiment (line 375-377).

The numbers of alimentary canals and salivary glands for FISH analysis are mentioned at the end of the paragraph: ‘five to ten samples were detected by FISH for each tissue’ in line 398.

18) L369. *Based on the description of the methods interpreted above, it would be *impossible* to have 30 ACP adults with which to do the plant inoculation experiments. I don’t know how I will be able to interpret these results given this consideration. I think I will have to consider them invalid.*

This was perhaps interpreted wrong. As outlined in the response above, we repeated enough

parallel groups to get the probed-but-survived 5th instar nymphs of ACP. Then 30 of them were randomly selected to inoculate new citrus shoots. Now the description is in line 401-402.

19) L403-4. *What is meant by 'relative titers'? Relative to what? The beta-actin gene? Please be more clear about this. I would like to see some more detail about this 'method 2(-delta delta)'. I realize that a reference is given and I presume that the method is correct, but just a summary of the methods would be helpful.*

We have sought to revise the sentence and added the housekeeping genes for CLAs in *Tamarixia* and ACP, respectively. The calculation method of $2^{-\Delta\Delta Ct}$ is detailed in the reference of Livak and Schmittgen 2001 and has been cited many times in scientific research. The value can be directly calculated in the qPCR machine (Bio-Rad). The description is in line 434-436.

20) L405. *The way I understand it, ANOVA is done on a replication level of 3, which is unusually low. I think it may be fine but it would be nice for readers to be assured that the assumptions of ANOVA were met by the data.*

We thank the reviewer for this query. The data were subjected to analysis of variance (ANOVA) when there were more than 3 treatments in the experiment; two-sample comparisons were performed by paired t-test (Fig. 1B). We have revised the sentence in the Data Analysis section in line 436-439.

Results:

21) Fig. 1. *I'm not quite sure what is meant by 'relative titer . . .'* Anyway, the finding of zero in eggs is good; it somewhat makes up for the error of not having a negative control for the parasitoids (noted above). You should be sure to mention this in the text (L90 – 97); I suggest writing something like, 'as expected, levels were zero in eggs . . .

Taking CLAs, ACP and *Tamarixia* as examples, the relative titers of CLAs in ACP and *Tamarixia*, are the relative amount/fold of CLAs in comparison to the expression levels of the housekeeping genes, *18S rRNA* of *T. radiata* and β -actin for ACP. We don't need to know the true amount of CLAs in the insects, since the expression levels of housekeeping genes are consistent. Taking the expression level of housekeeping genes as the references, we can determine how the CLAs amount changes before and after inoculation.

In regards to CLAs in eggs, we have revised the sentence to now read: "As expected, CLAs was not detected in eggs. CLAs titers in the 3rd and 4th instar larvae were significantly higher than those of 1st or 2nd instar larvae, pupae and adults (Figure 1 A)" in line 105-107.

22) Fig. 2 legend. *I believe you mean 'ovipositor' and not 'poison needle' (also to match the text).*

Thanks for highlighting this. We have corrected it in the legend of Figure 2.

23) L108-109. *I like the pictures in Fig. 3, but I think it's a bit overstating the results to say that they show spread from the mouthparts of the abdomen. This statement seems to come from one photo in which it's not clear what side of the larva has the mouth vs. the anus. If this statement is based upon more data please indicate that.*

We have revised the appropriate sentence to read as follows: ‘Over the course of development of *T. radiata* larvae, CLAs extended its distribution gradually, and as a result, CLAs had the strongest fluorescent signal in the middle of the abdomen of 4th instar larvae (Figure 2E)’ in line 120-123 .

And one thing need to mention, due to the limitation of Figures, we reduced the number of figure from 14 to 10 in the current version, and the others were put in online supplementary files.

24) *More importantly – we are not told (i) how many individuals were investigated for these analyses, or (ii) of those, how many showed positive results, or (iii) if the description of the FISH results are from more than one individual, how consistent the results were from individual to individual. I have the same comment for all of the FISH results.*

We have added this information into the M&M and Results sections. About 17-20 samples for the 1st to 4th instar nymphs were FISH analyzed (line 123-124), and all the CLAs positive individuals in each instar were consistent to each other with their CLAs distributions.

25) *L115. Why is the DuFour’s gland mentioned here for the first time? It was not mentioned in the methods. If you really did find this gland, please explain how you identified it as such.*

We aimed to FISH analysis the CLAs in the reproductive and venomous organs of *Tamarixia radiata*. Therefore, we did not detect the relative titer of CLAs in DuFour’s gland in the experiment leading to Figure 1. However, the DuFour’s glands are connected to the poison sac, it is the tissue extension of the poison sac. Thus, we also now show the FISH finding in Figure 3. To assist in understanding, please see figure below.

26) *Section starting on L116. Please recall that we were told in the methods (L327-8) that only parasitoids testing positive for CLAs would be used for these analyses. We need to be told what the proportion positive was (you can provide that information in the methods section or here, but you need to provide it).*

We have added the information into the M&M Maternal transmission of CLAs section: 58/60 and 56/60 for *T. radiata* females and males respectively were CLAs-infected, in line 354.

27) *L117-120. This sentence is too long and difficult to understand. I had to read it multiple times to understand it. Please work on clarifying this. For instance, the titer was higher in eggs than what? I would suggest breaking this sentence up into 2 or 3.*

We have sought to revise these sentences to make them clearer in line 134-137.

28) *Fig. 6. I'm not sure how the results in Fig. 6 can be compared to the results in Fig. 1. Both say 'relative titers', but only in Fig. 6 are we told about the fold change. It would be great to have all of the figures showing the qPCR results displayed in a way that they can be easily compared to one another (maybe that's how this was done but if so this needs to be made more clear).*

Figure 1 shows the relative titers and their fold changes in *T. radiata* F0 generation, which was developed from the CLas-infected ACP hosts. Figure 6 shows the same items of *T. radiata* F1 generation that was the progeny of F0 developed from the CLas-free ACP hosts. The relative titers of CLas in different charts is calculated in the same way, we have unified the description of the legends in the text. "Error bars represent the mean value \pm SE of three replicates Columns with the same letter represent means with no significant difference at $P < 0.05$." The original Figure 6 is now the Figure 4.

29) *Section starting on L128. In this section we need to be told the percentage of surviving nymphs that were positive for CLas. Also – the proportion of parasitoids testing positive (from L344).*

We have added the requested information in the text: approximately 95% of the *T. radiata* developed from CLas-infected ACP nymphs were CLas positive; while the survived ACP nymphs from the wasp probing were 100% CLas positive, in line 371-374.

30) *L136. Please don't call the parasitoid a 'phoretic vector' at this point. This is the hypothesis you are testing; you are treating it like a conclusion.*

We have rewritten these sentences in the text (line 147-152).

31) *Section starting on L151. As for all of the other results, there is no indication of sample size here.*

We clarified the sample size in the M&M: ".....30 more of these adults were randomly selected to inoculate on fresh shoots of CLas-free citrus...." in line 401-402.

Discussion:

32) *Based upon my skepticism of the results (outlined above) I disagree with parts of the Discussion section.*

We appreciate the skepticism and as a result, we have revised the manuscript carefully according to your valuable comments. We trust that our responses above (along with revisions made as requested by the other reviewers) will enhance the quality of this manuscript.

33) *I do agree with the recommendation that CLas-free parasitoids be used in releases since I do believe that you have shown the potential for vectoring under the special circumstance of hosts surviving an attack.*

We appreciate the supportive comment.

Reviewer #3 (Remarks to the Author):

In their manuscript, the team of Prof. Qiu present a new route for plant pathogen spread. Parasitoids that interacted with herbivore hosts that vector plant pathogens, became carries of the plant pathogen. By visiting herbivores on other trees, the parasitoid may vector the plant disease via parasitizing or probing herbivores feeding on this tree. Overall, I find that the authors provide convincing evidence of pathogen being vectored by parasitoids, with its most critical experiment confirming pathogen infection in plants after herbivores that interacted with parasitoids fed from these plants. I have no technical reservations, the study seems to be carefully conducted. I do have a number of suggestions for this manuscript to connect to a broader readership:

1) *The introduction focusses on a single plant-pathogen-herbivore-parasitoid system. The broader context of disease spread by insects and how these interactions may be affected by parasitoids could make the paper more interesting to readers that are not working on citrus pests.*

We thank the reviewer for this valuable suggestion. We have added the following to the end of the introduction to extend the context to other related research systems: “Also, our findings demonstrate that the potential negative effect of parasitoid-based horizontal transmission of plant pathogens should carefully appraise in other tri-trophic systems such as plant-aphid/whitefly/mealybug-parasitoid interactions” in line 100-101.

2) *Following this suggestion, I miss a more general picture of what is already known from the role of parasitoids in herbivore vectored plant diseases. Several studies have evaluated how parasitoids as biocontrol agents may affect disease spread and biocontrol may not yield the desired effect (or even promote disease spread). Parasitoids/predators may affect herbivore behaviour and cause increase in disease spread by increasing herbivore movement among plants. Plant pathogens may affect recruitment of parasitoids. Herbivores infected with plant diseases may affect performance of parasitoids. To me the results of this study still add a novel component to insect vectored diseases, but when placing those results in the context I suggest here would increase the value of the findings. The connection to broader literature on the role of parasitoids in plant disease spread could be discussed in both the introduction and discussion.*

As discussed in the second paragraph of the Discussion section (current revised version), a few examples have been reported in which parasitoids and mites can transmit bacterial symbionts horizontally between different insect hosts. However, our study is the first to reveal that a parasitoid can transmit a plant bacterial pathogen.

By line number:

3) *Line 35: The rational of using parasitoids.... Has never been evaluated. This is not entirely true in the context that I sketch in my general feedback. It is known that parasitoids may have unexpected negative effects by increasing plant disease spread. It is not know that parasitoids themselves may vector the disease (line 36).*

We do agree with the reviewer’s comments. Here we just focus on the *Tamarixia* parasitoid in citrus psyllid biological control (line 32-33). Its potential risk has never been evaluated, and

this is why we undertook the current study.

4) *Line 36: Change to “..we report that the most dominant parasitoid of ACP (Tamarixia radiata) could vector..”*

We have revised the sentence as suggested in line 33-35.

5) *Line 41: This sentence is unclear*

We have sought to revised much of the abstract in line with all the reviewer’s comments, and reduced the length of Abstract to 150 words.

6) *Line 48: this sentence is long and complex. I also suggest to delete the “for the first time” here. This is already stated in line 36.*

We have deleted “for the first time” and revised the sentence as suggested in line 37.

7) *Line 91 and elsewhere: The results section could be condensed by using a more direct writing style. Remove “Results showed that the titers...” and start right away with “The titers...”*

We appreciate this comment. Other reviewers have also raised similar issue. To this end we have sought to revise the text appropriately.

8) *Line 94: Here you highlight that parasitoid males have high titers of the plant pathogen. This aspect is not discussed and the consequences for disease spread could be added to the discussion (or I missed this?).*

We thank the reviewer for this valuable suggestion. We have added 5 lines into the Discussion section dealing with this. The feeding behavior of male *Tamarixia* parasitoid also has the potential risk to transmit CLAs among ACP individuals (line 213-217).

9) *Line 98: add s to adult*

We have corrected this, in line 112.

10) *Line 111-115: this paragraph describes similar results as the previous, but using a different technique. I suggest to integrate the two paragraphs*

Thanks for the comments, the previous paragraph focuses on the titers of CLAs from the qPCR investigation. This paragraph in question focuses on the distribution of CLAs from the FISH analysis. We believe it is easier to read and follow if the two experimental sections are described separately.

11) *Line 122: change transmitted to transmit*

We have changed as requested, in line 137.

12) *Line 182: entosymbiont should be endosymbiont*

We have corrected the spelling mistake, in line 198.

13) *Line 266: how were the individual marked?*

We have deleted the words “..and marked” from this sentence, in line 290.

REVIEWERS' COMMENTS:

Reviewer #2 (Remarks to the Author):

I am happy to see that the experiments were done with sufficient sample size and I can now much better understand what was done. I am disappointed though, that the authors don't seem to care that their scenario depends on the hosts surviving parasitism, and that their assays were done in test tubes which might improve the likelihood of host individuals surviving parasitism. Their only response was that such survival might be higher in the field. OK but it might be lower too right? To me their answer seemed non-sensical, but I may have misunderstood it because of the language barrier. I was also disappointed to see that this issue was not raised at all in the Discussion section - the authors simply state that they showed the potential for infection, period. They should say something like, 'in the unlikely event that hosts survive the parasitism process, infection is possible' and also include what is known about the likelihood that hosts survive parasitism (note that their test-tube assay does not provide information on this point).

In my opinion, a data set from the field or a more realistic lab study should be done to estimate the rate of survival, and this should be highlighted as an important part of this interaction, rather than swept under the rug.

Dear editor,

We are so grateful to you and all the reviewers in this second round review. We have carefully considered the suggestion from you and the reviewer 2#, the following is our response about how we have revised the manuscript according to your suggestions. All authors are agree with our revision in current version.

Reviewer #2 (Remarks to the Author):

I am happy to see that the experiments were done with sufficient sample size and I can now much better understand what was done. I am disappointed though, that the authors don't seem to care that their scenario depends on the hosts surviving parasitism, and that their assays were done in test tubes which might improve the likelihood of host individuals surviving parasitism. Their only response was that such survival might be higher in the field. OK but it might be lower too right? To me their answer seemed non-sensical, but I may have misunderstood it because of the language barrier. I was also disappointed to see that this issue was not raised at all in the Discussion section - the authors simply state that they showed the potential for infection, period. They should say something like, 'in the unlikely event that hosts survive the parasitism process, infection is possible' and also include what is known about the likelihood that hosts survive parasitism (note that their test-tube assay does not provide information on this point).

In my opinion, a data set from the field or a more realistic lab study should be done to estimate the rate of survival, and this should be highlighted as an important part of this interaction, rather than swept under the rug.

Response:

After collecting CLas-parasitic wasps, they would first undergo starvation treatment for 1-2 hours to ensure that the parasitic wasps could quickly probe the nymph of citrus psyllid, which intensified the damage of the parasitic wasps to the nymph of citrus psyllid. Therefore, compared with the wild environment, the survival rate of psyllid nymph after being probed in this experiment was lower. Thus, combining the space factor, we think the survival rate of probe checked ACP would be higher in field than in our tube experiment.

Limited by time, we did not conduct field trials and could not provide further experimental data, the aim of our study was to reveal the potential risks of a parasitic wasp, from a qualitative perspective.

We have clearly stated that these experiments were performed under laboratory conditions and may not be representative of the full suite of biotic and abiotic factors present in field conditions in the Discussion part, line 255-257.